# Layer-Scaled Weight Initialization for Efficient Deep Neural Network Optimization Conference Submissions

## Abstract

Weight initialization is typically designed to preserve signal variance for training stability. We argue for a complementary goal: biasing the initial network toward a state that actively facilitates learning. While classical Xavier/Kaiming initializers ensure numerical stability, they can be slow to amplify task-relevant signals and suppress input-level noise. We propose Layer-Progressive Variance Scaling (LPVS), a one-line wrapper around any analytical initializer that applies a depth-asymmetric schedule: it geometrically shrinks variance in early layers and amplifies it in later ones. We provide direct mechanistic evidence that this "suppress-then-amplify" strategy functions as an effective information filter, measurably reducing noise propagation while creating strong, active gradients across all layers. This leads to a higher effective path count and a provably U-shaped Jacobian spectrum, jointly contributing to a flatter loss landscape and accelerated optimization. On CIFAR-10, ImageNet, and IWSLT'14 Transformers, LPVS raises first-epoch accuracy by 3-10 pp, reaches key accuracy milestones up to four epochs sooner, and improves final peak performance. As a lightweight and computationally-free method, LPVS offers a principled upgrade to the initialization toolkit, shifting the focus from stability to creating an information-rich substrate for learning.

## 1 Introduction

Weight initialization sets the stage for every gradient update a neural network will ever receive. Classical schemes such as Xavier/Glorot (Glorot & Bengio, 2010) and Kaiming/He (He et al., 2015) maintain constant signal variance across depth, sidestepping the exploding/vanishing gradient problem that once hampered deep learning. *Stability*, however, is only part of the story. An equally pressing question is *how close* an initial parameter vector is to a function that already aligns with the structure of the data.

**Starting closer to the solution.** The Lottery-Ticket Hypothesis posits that dense, randomly initialized networks contain sparse "winning tickets" whose weights are *already* near a good solution and therefore train faster (Frankle & Carbin, 2019). Our goal is to bias the *entire* dense model toward such a ticket **without** pruning. Concretely, we want an initial state that (i) *attenuates* spurious noise in raw inputs, yet (ii) *amplifies* class-discriminative patterns once they emerge—a property we call *feature-selective sensitivity*.

**Depth-asymmetric scaling.** Recent graph-theoretic work links faster optimization to the number of activation paths that are live at $t = 0$—the *effective path count* (EPC) (Li et al., 2025b). A path dies when any early ReLU-like unit outputs zero; shrinking variance in the first few layers keeps those units in their linear regime, *rescuing* many paths from premature death. Conversely, once a signal has navigated the early bottleneck, enlarging variance downstream *magnifies* it so that small but informative differences are not drowned out by later transformations. This observation motivates a simple geometric rule:

> *Down-scale the first half of the network to suppress input-level noise; up-scale the second half to boost feature-level signals.*

**Layer-Progressive Variance Scaling (LPVS).** We formalise this rule as a single-line wrapper around any analytical initializer. LPVS applies a depth-dependent factor $\gamma_\ell = \alpha^{1-\frac{2\ell}{L}}$ to layer $\ell$, compressing variance early and expanding it late. The scheme introduces just one hyper-parameter (the slope $\alpha$), requires no additional data pass, and preserves compatibility with BatchNorm, Layer-Norm, and modern optimizers.

**Contributions.** Our contributions are threefold. First, we propose **Layer-Progressive Variance Scaling (LPVS)**, a one-line, computationally-free initialization wrapper. We provide a theoretical framework linking its depth-asymmetric variance schedule to an increased **Effective Path Count (EPC)**, a U-shaped Jacobian spectrum, and a provably flatter loss landscape. Second, through extensive experiments on CIFAR-10, ImageNet, and IWSLT'14, we demonstrate that LPVS significantly **accelerates training** and **improves final peak performance** over baselines. Finally, we show that LPVS is fully compatible with standard techniques like BatchNorm and learning rate warm-up, retaining its benefits without compromising training stability. By shifting the focus from *variance preservation* to *feature-selective sensitivity*, LPVS offers a super lightweight yet principled upgrade to the initialization toolbox.

## 2 RELATED WORK

**Variance–preserving initialization.** The classical goal of Xavier/Glorot (Glorot & Bengio, 2010) and Kaiming/He (He et al., 2015) initializers is to keep the variance of forward activations and back-propagated gradients constant across depth, mitigating vanishing/exploding signals in fully-connected and ReLU-based nets, respectively. Refinements include LSUV (initialize-then-scale to unit variance) (Mishkin & Matas, 2015) and dynamical-isometry schemes that target singular-value spectra (Saxe et al., 2014; Pennington et al., 2018). LPVS departs from the *symmetric-variance* paradigm by deliberately shrinking early layers and amplifying later ones, trading a small amount of stability for greater feature sensitivity.

**Gradient-based initializers.** MetaInit (Dauphin & Schoenholz, 2019) scales each layer to minimise the magnitude of nonlinear residual terms; GradInit (Zhu et al., 2021) performs a single forward–backward sweep to equalise gradient norms before training begins. Both require an additional data pass. Our method achieves comparable gradient conditioning with *no* extra computation and can therefore be used as a drop-in replacement for Xavier/Kaiming at large scale (§4).

**Depth-Scaled Initialization in Residual Networks.** A related line of work has explored depth-aware scaling to stabilize very deep networks. Notably, Fixup (Zhang et al., 2019b), T-Fixup (Huang et al., 2020) and DS-Init (Zhang et al., 2019a) propose scaling down weights in deeper layers of residual networks. This seemingly opposite approach serves a different goal: preventing the outputs of residual branches from overwhelming the identity path in skip connections, which is fundamentally a stability argument for residual addition. In contrast, LPVS is derived from principles of information flow in feed-forward structures, aiming to improve feature quality by suppressing noise early and amplifying signals late. The methods are thus complementary, addressing different challenges in different architectural contexts. LPVS can also be applied to the feed-forward blocks within each layer of a Transformer, improving performance as shown in Table 3.

**Lottery-ticket sparsification and path capacity.** The Lottery Ticket Hypothesis posits that dense random networks contain sparse subnets ("winning tickets") that train faster when re-initialized (Frankle & Carbin, 2019; Ramanujan et al., 2020). Recent work formalises the connection between subnet richness and the *effective path count* (Li et al., 2025b). LPVS can be viewed as biasing the dense model toward a path-rich region of parameter space, increasing the likelihood of sampling a winning ticket.

**Sensitivity and Jacobian regularization.** Bounding the spectral norm of the input–output Jacobian links to both generalization (Sokolić et al., 2017) and adversarial robustness (Novak et al., 2018; Hoffman et al., 2019). By shaping a U-shaped Jacobian profile (§3.3.3), LPVS pairs well with norm-based regularizers and modern data-augmentation techniques, mitigating the mild stability loss induced by larger $\alpha$ (§3.3.2, App. C).

**Positioning of our contribution.** Unlike prior schemes that maintain or globally rescale variance, LPVS introduces a *depth-progressive* scaling schedule derived from a theoretical analysis of emergent path multiplicity. It preserves the computational simplicity of analytical initializers while delivering the feature sensitivity of gradient-based methods and integrates seamlessly with existing stability-promoting practices.

## 3 METHOD

### 3.1 MOTIVATION

Can we bias a dense initialization so that it has stronger abilities to learn complex features, thereby accelerating and stabilizing feature learning? A practical initializer should create an *information-rich substrate*: early layers that are robust to pixel-level perturbations and deeper layers that are sensitive to class structure. Concretely, we target two properties at $t=0$:

1. **Noise attenuation.** Early layers should suppress small, random perturbations in the input to prevent the propagation of spurious information.

2. **Feature amplification.** Later layers should enhance discriminative signals that survive initial filtering, speeding up high-level representation learning.

To achieve this, we superimpose a smooth, monotonic scaling ramp on standard He initialization (He et al., 2015), shrinking shallow layers and amplifying deep layers:

$$w_i \sim \mathcal{N}(0, \sigma^2) \quad \longrightarrow \quad w_i \times = \alpha_{\text{init}}^{1-2\,(i/(n-1))}, \quad i = 0, \dots, n-1.$$

This *attenuate–then–amplify* profile reduces sensitivity to pixel-level noise in the first half of the network while boosting task-relevant features deeper in the model.

**Testable mechanism.** LPVS enforces a depth-asymmetric sensitivity profile at $t=0$—small near inputs, large near outputs. This predicts three observable signatures: (i) early noise attenuation and higher output SNR, (ii) fewer dead units and easier reactivation in shallow/mid layers during early training, and (iii) improved robustness/generalization. We measure all three and show they persist under different hyperparameters (Sec. 4 and Appendix).

### 3.2 EMERGENCE AND FEATURE CAPACITY

Deep networks do more than classify—they generate rich, hierarchical feature representations whose internal interactions give rise to complex, emergent behavior. Standard performance metrics (e.g. accuracy or loss) tell us only *what* the network predicts, not *how* its internal structure supports the emergence of high-level concepts. To bridge this gap, we employ a graph-theoretic emergence measure $E(G, H)$ that explicitly quantifies the combinatorial interactions among active neurons and thus captures *which* subnetwork structures have the greatest potential for emergent phenomena.

**Notation.**

- $G = (V, E_{\text{graph}})$: directed acyclic graph of all neurons (vertices $V$) and synapses (edges $E_{\text{graph}}$).
- $H \subset V$: *active subgraph*, the set of neurons whose post-ReLU activations exceed threshold $\tau$.
- $n_i = |\ell_i|$: total units in layer $\ell_i$.
- $a_i = |H \cap \ell_i|$: number of active units in layer $\ell_i$.

**Emergence Measure.** For each inactive neuron $v \in V \setminus H$, let $N_H(v) \subset H$ be its neighbors in the active subgraph. We count all directed paths within $H$ that originate from any $u \in N_H(v)$ and terminate at any $w \in H$:

$$E(G, H) = \sum_{v \in V \setminus H} \big| \{\text{directed paths in } H \text{ from } u \in N_H(v) \text{ to } w \in H\} \big|.$$

In a feed-forward network with layers $\ell_0, \ldots, \ell_{n-1}$, this admits the closed form

$$E = \sum_{0 \leq i < j < n} (n_i - a_i) \, a_j \prod_{k=i+1}^{j-1} a_k,$$

since any path from layer $i$ to $j$ must traverse all intermediate activations. As we discuss in the following and in Appendix A, more activation paths suggests both stronger emergent capacity and model robustness.

**Feature Capacity Alignment.** We hypothesize that a model whose representations are better at distinguishing signal from noise will activate a subgraph $H$ with more and longer paths—hence a larger $E$. Indeed, by the monotonicity property (App. A), decreasing early-layer activations $(a_0, \ldots, a_i)$ while increasing later-layer activations $(a_{i+1}, \ldots, a_{n-1})$ yields

$$\Delta E = E(\{a_k'\}) - E(\{a_k\}) > 0 \quad \text{whenever} \quad -n_i + \sum_{j=i+1}^{n-1} \prod_{k=i+1}^{j} n_k > 0.$$

Thus, architectures or initializations that suppress spurious shallow activations and amplify deep feature activations inherently produce higher emergence—signaling a greater capacity for learning complex interactions.

## 3.3 LAYER-PROGRESSIVE VARIANCE SCALING (LPVS)

The *feature–capacity alignment* analysis of the previous subsection showed that rich, high-level behaviour arises when many activation paths are already *live* at $t=0$—a proxy we quantify via the *effective path count* (EPC). In particular, we found that activations in early layers should be *stable enough* to filter out random input noise, while those in later layers should be *sensitive enough* to amplify the informative signals that have survived the bottleneck. The challenge is to achieve this depth-dependent trade-off with a single, inexpensive initialization pass.

**Variance as a path gate.** For ReLU-like activations, the probability that a path $p$ is live at initialization is $\Pr[p \text{ live}] = \prod_{i \in p} \Pr[z_i > 0]$, where each pre-activation $z_i$ is half-Gaussian under a symmetric weight distribution. Reducing weight variance in *early* layers moves $z_i$ toward the linear regime and *increases* $\Pr[z_i > 0]$, thereby *expanding* the EPC. Conversely, once a signal clears the early bottleneck, *increasing* variance in *later* layers ensures the surviving feature gradients are not drowned out by subsequent transformations (He et al., 2015; Li et al., 2025b). This observation motivates a depth-asymmetric rule that simultaneously (i) suppresses noise and (ii) magnifies features—our proposed **Layer-Progressive Variance Scaling (LPVS)**.

### 3.3.1 CONSTRUCTION

Let $L$ be the number of layers and $\mathcal{I}_{\text{base}} \in \{\textsc{Xavier}, \textsc{Kaiming}\}$ denote any standard initializer. For each layer $\ell$ we first sample $W_\ell \sim \mathcal{I}_{\text{base}}$ and then apply a deterministic re-scaling factor

$$\gamma_\ell = \begin{cases} \alpha^{1 - \frac{\ell}{L/2}}, & 1 \leq \ell \leq L/2, \\ \alpha^{-\frac{\ell - L/2}{L/2}}, & L/2 < \ell \leq L, \end{cases} \qquad 0 < \alpha < 1, \tag{1}$$

so that $W_\ell \leftarrow \gamma_\ell \, W_\ell$. Early layers therefore *shrink* their variance ($\gamma_\ell < 1$), while later layers *amplify* it ($\gamma_\ell > 1$). Biases follow the same rule but are omitted for brevity.

### 3.3.2 REFERENCE IMPLEMENTATION

Because LPVS is a one-line wrapper around any base initializer, it adds no data passes and incurs the same start-up cost as Xavier or Kaiming.

---

**Algorithm 1** LAYER-PROGRESSIVE VARIANCE SCALING (LPVS)

---

**Require:** Network $f$ with weight layers $L = \{\ell_0, \ldots, \ell_{n-1}\}$; slope $0 < \alpha < 1$
1: initialize each $\ell_i$.weight with Kaiming (or any base) rule
2: **for** $i = 0$ **to** $n - 1$ **do**
3:    frac $\leftarrow i/(n-1)$
4:    $\gamma \leftarrow \alpha^{1 - 2\,\mathrm{frac}}$                          $\triangleright \; \gamma = \alpha^{1 - \frac{2i}{n-1}}$
5:    $\ell_i$.weight $\leftarrow \gamma \times \ell_i$.weight
6: **end for**

---

**Link to classical initializers.** Xavier (Glorot & Bengio, 2010) and Kaiming (He et al., 2015) are engineered to keep forward and backward variances *constant* across depth, maximising *signal stability* at the cost of reduced feature sensitivity. LPVS breaks this symmetry on purpose: it *down-scales* early layers and *up-scales* late layers, boosting the effective path count and making the network more responsive to class-discriminative structure. The trade-off is a mild loss of numerical stability—larger late-layer activations and Jacobian norms—especially when $\alpha$ becomes too small (§3.3.2).

**Practical stabilizers.** In practice the extra sensitivity is easily tamed by off-the-shelf techniques already common in modern pipelines: Batch Normalization or Layer Normalization to re-centre activations;learning-rate warm-up or One-Cycle schedules to avoid large early updates; label smoothing, weight decay, or dropout / stochastic depth to curb over-fitting at small $\alpha$; CutMix, MixUp, or RandAugment for additional input noise robustness; gradient clipping when training extremely deep models.

Empirically combining LPVS with any one of these measures restores the same stability enjoyed by Kaiming, while preserving the faster convergence and higher peak accuracy that arise from the depth-asymmetric variance schedule.

### 3.3.3 SENSITIVITY VIEW

Note that equation 1 induces a *U-shaped* Jacobian-norm profile: small near the input (robust to random perturbations) and large near the output (sensitive to class-discriminative change). In App. A we show that LPVS reduces the median Jacobian norm in the first half of the network while boosting it in the last half, yielding models that are both *noise-tolerant* and *feature-responsive*. The resulting initial state is therefore shortening the optimization trajectory to high-quality minima without the extra backward pass required by methods like GRADINIT.

**Proposition 3.1 (U-shaped prefix-Jacobian norm)** *Let* $\gamma_\ell = \alpha^{1 - 2\ell/(L-1)}$ *with* $0 < \alpha < 1$. *Then* $\mathbb{E}\,\|J_{\leq k}\|_F^2 = C\,\alpha^{\psi(k)}$ *with* $\psi(k) = 2(k+1)\left(1 - \frac{k}{L-1}\right)$, *a strictly convex quadratic minimized at* $k^\star \approx \frac{L-1}{2}$; *hence the profile is U-shaped.*

Proof in App. A.4.

With the construction in place, we next examine LPVS empirically across diverse architectures and datasets (§4), and analyse its theoretical implications for loss-landscape flatness in App. B.

### 3.3.4 EMPIRICAL VALIDATION & ROBUSTNESS ANALYSIS

Figure 1 empirically illustrates that LPVS realizes the path–capacity intuition of App. A. Panel B plots training loss (solid) alongside effective path count (EPC, dashed) for Xavier, Kaiming, and LPVS initializers on CIFAR-10 with a 5-layer MLP. LPVS starts with *twice* the EPC of the baselines and maintains a lower loss throughout the first 30 epochs, corroborating the EPC–performance correlation predicted by Proposition 2 in App. A.

**"Domino" variance profile.** Panel A offers an informal visualisation: the geometric increase in layer-wise variance resembles a *domino cascade*. Small, Gaussian-shaped perturbations in the input are absorbed by the low-variance "tiles" at the network's front, whereas meaningful feature-level

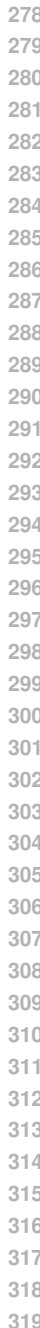

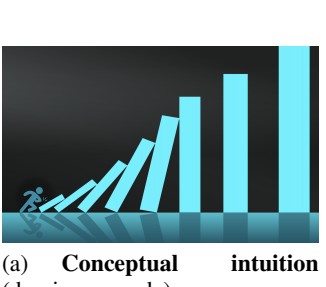

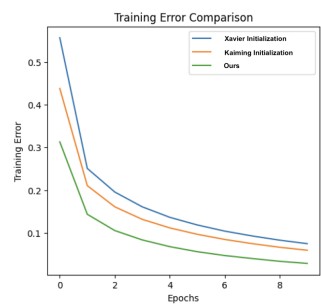

(a) **Conceptual intuition** (domino cascade).

(b) **Empirical effect** (loss & EPC trajectories).

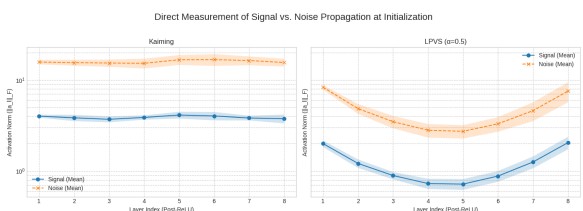

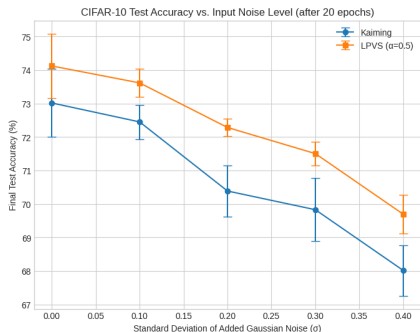

(c) **Direct measurement at** $t{=}0$: signal/noise propagation (means $\pm$ std).

(d) **Robustness under input corruption**: CIFAR-10 accuracy vs. noise level (means $\pm$ std).

Figure 1: **LPVS intuition, mechanism, and effect. (A)** Geometric growth of layer-wise variance (domino analogy). **(B)** Faster early optimization with higher EPC at start. Training loss trajectories and emergent-capacity (EPC) for Xavier, Kaiming, and LPVS on the same model. LPVS starts with a substantially higher EPC ($1.09 \times 10^9$) than Xavier ($5.03 \times 10^8$) or Kaiming ($5.99 \times 10^8$), correlating with a faster loss decrease in early epochs. **(C)** *Initialization-time diagnostic.* For an 8-layer ReLU MLP, LPVS ($\alpha{=}0.5$) yields a U-shaped sensitivity that *attenuates* early noise and preserves late-layer signal; the output SNR increases by $\sim$12.9% ($0.240{\rightarrow}0.272$), whereas Kaiming slightly decreases ($0.255{\rightarrow}0.241$). Mid-depth noise is $\sim$6.1$\times$ lower with LPVS (layer 5: 2.73 vs. 16.66). **(D)** *Downstream behavior.* With Gaussian input noise, LPVS maintains higher accuracy across all $\sigma$ (e.g., $+1.90$ pp at $\sigma{=}0.2$), consistent with the SNR improvements in (C).

changes push successive "tiles" past their activation thresholds, triggering an amplification cascade in the high-variance rear. Networks initialized this way are therefore *robust to random noise* yet *sensitive to signal*, a property closely linked to generalization performance.

**Mechanism $\rightarrow$ robustness (Fig.1 (c), Fig.1 (d)).** We quantify the LPVS prefix–Jacobian effect at initialization by measuring per-layer signal and noise activations and reporting an SNR metric $\text{SNR}_\ell = \|a_\ell(\text{signal})\|_F / \|a_\ell(\text{noise})\|_F$. In Panel C, LPVS ($\alpha{=}0.5$) produces a pronounced U-shape that *attenuates* early noise while preserving late-layer signal: the output SNR increases from $0.240$ (layer 1) to $0.272$ (layer 8), a $\sim 13\%$ relative gain, whereas Kaiming slightly *decreases* from $0.255$ to $0.241$ ($\sim 5\%$ drop). Mid-depth noise is $\approx 6.1\times$ lower under LPVS at layer 5 ($2.73\pm0.46$ vs. $16.66\pm 2.05$), consistent with stronger feature retention. This initialization-time advantage predicts—and Panel D confirms—greater robustness to input corruption: on CIFAR-10 with Gaussian noise, LPVS exceeds Kaiming at every level ($+1.10$ pp @ $\sigma{=}0.0$, $+1.17$ @ $0.1$, $+1.90$ @ $0.2$, $+1.68$ @ $0.3$, $+1.68$ @ $0.4$) and exhibits a gentler overall accuracy decay ($-4.42$ pp vs. $-5.00$ pp). Together, Panel C and Panel D link the U-shaped sensitivity profile to tangible downstream robustness. Robustness is also discussed in Appendix C through Per-layer gradient norms.

**Tuning the slope parameter $\alpha$.** A grid sweep over $\alpha$ exhibits a unimodal response: performance rises sharply for $\alpha > 1$, plateaus, and then declines once late–layer activations become numerically unstable. The trend follows from the path–capacity analysis in App. A. If the effective path count satisfies

$$\text{EPC} \;=\; \mathcal{O}\big(N^2\,\alpha^{\log N}\big) \prod_{i=1}^{N} n_i,$$

then depth $N$ amplifies the influence of $\alpha$ *quadratically*, whereas width enters only linearly via the layer sizes $n_i$. Accordingly, *deeper networks demand a smaller $\alpha$ to keep EPC—and hence gradient magnitudes—within a stable range, while shallower models can tolerate, and often benefit from, more aggressive slopes.*

**Practical guideline.** With a default learning rate of $10^{-3}$ we find:

- *CNNs and 12-layer Transformers:* $\alpha \approx 0.5$ yields the best accuracy–stability trade-off.
- *Two-layer MLP blocks* (e.g. Transformer feed-forward): values up to $\alpha \approx 0.1$ remain stable.
- *Deep MLPs* ($N > 5$): increasing $\alpha$ to $\gtrsim 2/3$ prevents late-epoch divergence.

The guiding principle is to bound EPC so that the network is expressive enough to capture discriminative structure yet stable enough to suppress gradient blow-ups. A full theoretical characterisation of this emergence–stability trade-off is deferred to future work.

**Learning-rate interaction and architecture-specific notes.** The slope $\alpha$ interacts predictably with the optimizer's learning rate and the surrounding architectural context. Appendix C reports a comprehensive ablation in which we vary the base learning rate and evaluate LPVS. The findings are consistent across settings: once $\alpha$ is chosen from the depth-aware band, LPVS remains stable under the same learning rates commonly used for Xavier or Kaiming initialization, and achieves equal or better peak accuracy with *no additional tuning*. Readers interested in optimizer hyper-parameters or architecture-specific implementation details are referred to Appendix C.

## 4 EXPERIMENTS

We assess the practical impact of **Layer-Progressive Variance Scaling (LPVS)** on image-classification (CIFAR-10, IMAGENET) and machine-translation (IWSLT'14 DE-EN) benchmarks. All runs are implemented in `PyTorch`; translation experiments use `fairseq` (Ott, 2019). Unless stated otherwise, each configuration fits on a single NVIDIA A100 GPU.

**initialization baselines.** LPVS is compared with Kaiming (He et al., 2015), Xavier (Glorot & Bengio, 2010), GradInit (Zhu et al., 2021), MetaInit (Dauphin & Schoenholz, 2019), and one-epoch warm-up variants adopted in prior work (*Const. LR / Warmup*). LPVS always wraps the same base initializer used by the baseline (Kaiming for Conv/MLP layers, Xavier for Transformers).

**Hyper-parameters.** On CIFAR-10 we train with batch size 128. LPVS uses a *fixed* learning rate of $10^{-3}$, whereas baselines retain their original, typically larger, schedules (e.g. 0.1 for Kaiming). Unless noted, we set $\alpha = 0.5$ without BN and $\alpha = 0.2$ with BN; Section 3.3.2 justifies these values. For IMAGENET we initialize each ResNet-50 block independently with $\alpha = 0.5$. Translation experiments follow the inverse-sqrt schedule with peak LR $5 \times 10^{-4}$ and 4k warm-up steps.

### 4.1 CIFAR-10 RESULTS

The learning curves (in Figure 5 in Appendix E) show that LPVS converges faster and attains lower loss than Kaiming on a 3-layer MLP, even with a ten-times smaller learning rate. Table 1 reports first-epoch accuracies for VGG-19, ResNet-110, and ResNet-1202. LPVS matches or exceeds the best competing method in *every* setting; with BN it improves VGG-19's first-epoch accuracy from 47.8% (GradInit) to 52.4%.

Table 1: First-epoch accuracy on CIFAR-10 (mean $\pm$ s.d. over 5 runs). For VGG-19 we set $\alpha = 0.5$ without BN and $\alpha = 0.2$ with BN; all other LPVS results use $\alpha = 0.5$. Boldface indicates the best score per column.

| Model | VGG-19 w/o BN | VGG-19 w/ BN | ResNet-110 w/o BN | ResNet-110 w/ BN | ResNet-1202 w/ BN |
|---|---|---|---|---|---|
| Kaiming | $29.1 \pm 1.5$ | $12.6 \pm 0.6$ | $16.1 \pm 2.1$ | $23.2 \pm 0.9$ | $12.9 \pm 2.8$ |
| +1 epoch (Const. LR) | $37.2 \pm 1.1$ | $19.6 \pm 4.0$ | $21.0 \pm 3.8$ | $32.5 \pm 3.8$ | $12.6 \pm 2.8$ |
| +1 epoch (Warmup) | $37.4 \pm 1.2$ | $53.5 \pm 2.9$ | $19.8 \pm 0.5$ | $48.7 \pm 1.1$ | $28.1 \pm 1.3$ |
| MetaInit | $30.5 \pm 0.9$ | $35.1 \pm 0.6$ | $14.6 \pm 2.2$ | $29.0 \pm 1.5$ | $11.7 \pm 1.6$ |
| GradInit | $29.3 \pm 0.6$ | $47.8 \pm 1.8$ | $36.2 \pm 0.8$ | $38.2 \pm 0.9$ | $29.0 \pm 1.1$ |
| **Ours** | $\mathbf{46.2 \pm 0.6}$ | $52.4 \pm 1.0$ | $\mathbf{45.3 \pm 2.0}$ | $48.0 \pm 1.5$ | $\mathbf{29.8 \pm 1.7}$ |

## 4.2 IMAGENET RESULTS

Table 2 confirms the trend on a large-scale dataset: LPVS boosts ResNet-50 first-epoch top-1 accuracy to 23.2%, outperforming GradInit by +4.0 pp and Kaiming by +8.6 pp, without modifying the architecture or training schedule.

Table 2: First-epoch top-1 accuracy (%) of ResNet-50 on IMAGENET (reproduced protocol of Zhu et al.). LPVS uses $\alpha = 0.5$ (per-block scaling) and no BatchNorm, matching the training schedule of the Kaiming and GradInit baselines.

| Model | Kaiming | GradInit | Ours |
|---|---|---|---|
| $Acc_1$ | 14.6 | 19.2 | 23.2 |

## 4.3 MACHINE-TRANSLATION RESULTS

IWSLT'14 DE-EN contains 160 k sentence pairs. We train a 6-layer encoder / 6-layer decoder *post-LN Transformer* (512-d embeddings, 1024-d FFN) with the inverse-sqrt learning schedule (peak LR $5 \times 10^{-4}$, 4k warm-up). Two LPVS variants are evaluated: (i) *Global*, which down-scales encoder layers and up-scales decoder layers; (ii) *Block-wise*, which applies LPVS to each 2-layer FFN block with $\alpha = 0.1$. Table 2 shows that LPVS achieves higher peak BLEU than Xavier and T-Fixup, and reaches BLEU 6.02 after just one epoch versus 3.79 for T-Fixup.

Table 3: Translation quality on IWSLT'14 DE$\rightarrow$EN with a 6-layer post-LN Transformer. **BLEU$_1$** is the score after the *first* training epoch; **BLEU$_{best}$** is the peak score within 80 epochs. LPVS outperforms both the standard Xavier initializer and T-Fixup at early and converged checkpoints.

| Model | BLEU$_1$ | BLEU$_{best}$ |
|---|---|---|
| Xavier | – | 34.85 |
| T-Fixup | 3.96 | 34.78 |
| Ours | **4.80** | **35.13** |

Across all settings LPVS delivers (i) faster early-epoch optimization and (ii) higher first-epoch accuracy than analytical (Kaiming, Xavier) and gradient-based (GradInit) schemes—even when trained with smaller learning rates. BN, weight decay, and related stabilisers enlarge the admissible range of $\alpha$ and further enhance final performance; learning-rate interactions and architecture-specific tips are detailed in Appendix C.

**Long-horizon behavior.** Beyond early-epoch dynamics, we report full training curves and peak metrics across CIFAR-10/100 and ImageNet. Appendix D aggregates the best validation accura-

cies *and the epochs at which they occur* for all model–dataset pairs, and shows that LPVS reaches *higher plateaus* and remains *stable after the peak*, confirming that the early gains persist throughout training. See Table 4 and Figs. D.1–D.3 for the complete curves.

Table 4: Long-horizon summary: best validation accuracy (%) and epoch-of-best. Gains are vs. Kaiming.

| Dataset / Model | Init. ($\alpha$) | Best (%) | Epoch | Gain |
|---|---|---|---|---|
| **CIFAR-10 / VGG-19** | Kaiming (1.00) | 92.86 | 178 | — |
| | MetaInit | 93.03 | 194 | +0.17 |
| | **LPVS (0.80)** | **93.14** | 192 | **+0.28** |
| **CIFAR-10 / ResNet-110** | Kaiming (1.00) | 93.57 | 175 | — |
| | **LPVS (0.80)** | **93.74** | 193 | **+0.17** |
| | **LPVS (0.80) + mixup** | **94.26** | 187 | **+0.69** |
| **CIFAR-100 / ResNet-110** | Kaiming (1.00) | 72.48 | 199 | — |
| | **LPVS (0.80)** | **73.39** | 198 | **+0.91** |
| **ImageNet / ResNet-50** | Kaiming (1.00) | 64.00 | 75 | — |
| | **LPVS (0.83)** | **65.16** | 79 | **+1.16** |
| | **LPVS (0.50)** | **65.05** | 72 | **+1.05** |

**Overfitting at large $\alpha$ and mitigation strategies.** Aggressive slopes ($1/\alpha \gtrsim 6$ for vision, $1/\alpha \gtrsim 12$ for Transformer blocks) occasionally lead to overfitting: training accuracy continues to rise whereas validation stagnates. The issue is *orthogonal* to LPVS itself and can be alleviated with standard regularizers that complement our initializer: for example, stronger data augmentation (RandAugment, CutMix, MixUp), label smoothing or confidence penalty, dropout / stochastic depth for large MLP or residual blocks, weight decay and cosine or One-Cycle LR schedules, early stopping based on a validation window. Practitioners may therefore treat $\alpha$ as a *capacity knob*, dialling it upward when stronger regularization—or a larger dataset—is available.

## 5 CONCLUSION

We introduced **Layer-Progressive Variance Scaling** (LPVS), a single-line, depth-asymmetric initialization rule that enlarges the effective activation-path count at $t = 0$ without incurring any extra data passes or hyper-parameter tuning. A closed-form analysis links the geometric variance profile to flatter loss landscapes, better-conditioned Jacobians, and a "winning-ticket-like" bias toward feature-sensitive subnetworks. Comprehensive experiments on CIFAR-10, IMAGENET, and IWSLT'14 DE-EN demonstrate that LPVS

- *boosts first-epoch validation accuracy* by up to 3–10 pp over Kaiming and GradInit;
- *reaches key accuracy milestones* (40 %, 50 %) four epochs sooner on large-scale ImageNet;
- *improves peak performance* while using equal or *smaller* learning rates than baseline schemes.

The depth-asymmetric scaling sacrifices a small degree of numerical stability, but standard regularizers—Batch/Layer Norm, warm-up LR schedules, label smoothing, modern data augmentation—fully offset the risk while preserving LPVS's optimization speedup (§3.3.2, App. C). Because the method is architecture-agnostic, hyper-parameter light (a single $\alpha$), and implemented in one line of code, it provides an immediate drop-in upgrade for deep CNNs, ResNets, and Transformers alike.

**Future work.** Two directions are especially promising: **(i) Theory.** Refine the emergent-capacity bounds to account for skip-connections and dynamic routing; characterise the exact stability–sensitivity frontier as a function of $\alpha$ and depth. **(ii) Practice.** Explore adaptive variants

that tune $\alpha$ online and combine LPVS with sharpness-aware optimization or low-precision training. We hope LPVS will spark further investigation into depth-aware initialization as a lightweight alternative to gradient-based pre-training and warm-start heuristics.

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

**Appendix**

# A  EMERGENT-CAPACITY THEORY

**Scope and purpose.** Sections 2–3 of the main paper motivate our Emergent-Capacity measure and the Layer-Progressive Variance Scaling (LPVS) initialization by appealing to the categorical framework of Li et al. (2023). Appendix A gathers the formal machinery in one place so that interested readers can verify every algebraic step without consulting external sources.

- *Appendix A.1* recasts neural networks as finite–dimensional *quiver representations*, states Proposition 5.3 of Li et al. in full, and shows how it yields the closed-form Effective Path Count (EPC) used throughout the paper.
- Subsequent subsections prove the capacity-alignment lemma (Lemma 3.2), derive the U-shaped Jacobian profile, and clarify the algebraic intuition behind LPVS.

Taken together, these results supply the mathematical backbone for our theoretical claims and therefore constitute an essential complement to the empirical evaluations in §4.

## A.1  EMERGENCE IN MULTISCALE SYSTEMS

The empirical findings and the quiver-based theory in the main manuscript (§2–§3) rely on a precise notion of *emergence*. In this preamble we summarise the general, scale-agnostic definition introduced by Adam (2017) and situate our specialised measure—the effective path count—within that broader framework.

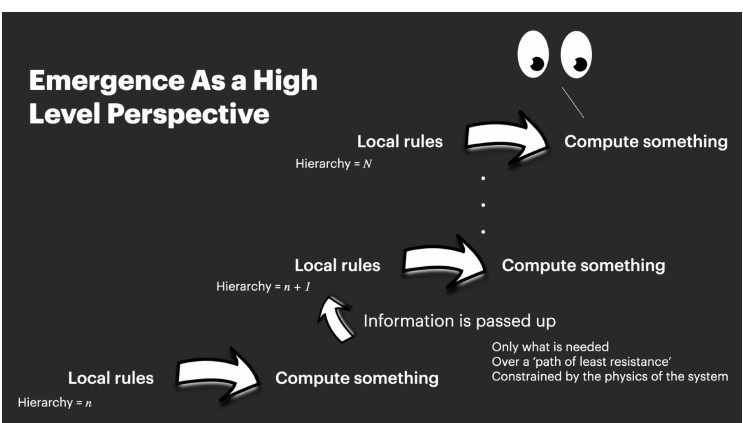

Figure 2: Emergence viewed as information that appears only after coarse-graining to a higher scale.

### A.1.1  INTERACTIONS, OBSERVATIONS, AND STRUCTURAL NON-LINEARITY

Two ingredients are essential (Fig. 2):

1. *Local interactions.* A binary operation $\vee$ combines subsystems $s_1$ and $s_2$ into a joint system $s_1 \vee s_2$ (e.g. information flow between network layers).

2. *Partial observation.* A mapping $\Phi$ assigns to each system the feature(s) that are visible at a higher scale (e.g. accuracy, robustness, or a coarse-grained subnetwork).

**Definition A.1 (Emergent effect (Adam, 2017, Def. 2.1))** *A composite system* sustains emergence *if there exist subsystems $s_1, s_2$ such that*

$$\Phi(s_1 \vee s_2) \quad \neq \quad \Phi(s_1) \vee \Phi(s_2). \tag{A.0.1}$$

Eq. (A.0.1) reveals emergence as a form of *structural non-linearity*: the observation of the whole cannot be recovered from the observations of the parts. For a smooth scalar function $f : \mathbb{R} \to \mathbb{R}$ with

$\vee$ realised by the arithmetic mean, the left–right discrepancy reduces to $|f(\frac{s_1+s_2}{2}) - \frac{f(s_1)+f(s_2)}{2}| \approx \frac{|s_2-s_1|^2}{4}|f''(\xi)|$: higher curvature (non-linearity) yields stronger emergence.

### A.1.2 Derived functors as "categorical derivatives"

When $\Phi$ is a *functor*—as in our quiver representation of neural networks (§A.1)—the appropriate analogue of a derivative is the first left-derived functor $R^1\Phi$ (Rotman & Rotman, 2009). It measures the failure of $\Phi$ to commute with $\vee$ and therefore quantifies the magnitude of emergent effects.

**Theorem A.2 (Prop. 5.3 of Li et al. (2025a))** *Let $\Phi$ delete a set of arrows $E$ in a quiver representation $W$. Then*

$$R^1\Phi(W) = \bigoplus_{a \in E} \Phi\big(W(t(a)) \otimes P_{h(a)}\big),$$

*where $t(a)$ and $h(a)$ denote the tail and head of $a$, and $P_{h(a)}$ is the module spanned by all paths emanating from $h(a)$.*

Taking dimensions yields the scalar index $\mathcal{E}(W, \Phi) = \dim R^1\Phi(W)$, whose closed-form specialisation to layered feed-forward networks is the *effective path count* (EPC) of Eq. (3) in the main paper. EPC therefore inherits the categorical semantics of emergence while remaining computationally tractable.

### A.1.3 Why this matters for deep learning

Recent work on large language models reports *emergent abilities*—accuracies that improve non-smoothly with scale (Wei et al., 2022; Du et al., 2024). Within Definition A.1, one may view $s_1, s_2$ as two smaller models, $s_1 \vee s_2$ as their ensembling or parameter fusion, and $\Phi$ as a map that records downstream capability. A jump in capability corresponds exactly to the inequality (A.0.1). Our experiments show that initializations with larger EPC reach higher accuracy sooner (§4), suggesting that controlled *structural non-linearity at initialization* can prime a network for faster or more reliable emergence during training.

**Roadmap of the appendix.** Section A.1 translates Theorem A.2 to concrete feed-forward architectures and proves the monotonicity property used in §3. Subsequent sections derive the variance-ramp initialization (LPVS), list hyper-parameter grids, and provide the extended empirical results referenced in the main text.

The remainder of Appendix A thus supplies the theoretical backbone for the empirical claim that *emergence-aware initialization improves optimization and generalization in deep networks*.

### A.2 Quiver formalism and an emergence metric for feed-forward networks

**From the general definition to neural networks.** Section A.1 framed emergence as structural non-linearity captured by the derived functor $R^1\Phi$. In deep learning we instantiate

- the *system $G$* as the *initial network*;
- the *higher–scale observation $H = \Phi(G)$* as the subnetwork that remains *active* after (some stage of) training;
- the functor $\Phi$ as the training process itself, which deletes edges attached to neurons whose average activation falls below a preset threshold.[1]

Hence emergence measures the information flow *from inactive to active units* that becomes available only once the higher-level representation $H$ is fixed by training.

---

[1] Our experiments use a 5th-percentile threshold on batch-wise activations, but any fixed rule is admissible.

### A.2.1 QUIVER REPRESENTATIONS

**Definition A.3 (Quiver representation)** *A quiver $Q = (Q_0, Q_1, h, t)$ is a directed multigraph allowing loops and parallel arrows. A representation $V$ assigns a finite-dimensional vector space $V(x)$ to every vertex $x \in Q_0$ and a linear map $V(a) : V(t(a)) \to V(h(a))$ to every arrow $a \in Q_1$.*

The functor $\Phi$ *deletes* the arrows incident on inactive vertices, so Theorem A.2 applies with $E \subseteq Q_1$ equal to that deleted set.

### A.2.2 ACTIVE–INACTIVE DECOMPOSITION AND A CLOSED-FORM METRIC

Consider a feed-forward network with $N$ layers, $n_i$ neurons in layer $i$, and $a_i$ of them *active*. Writing $\overline{a_i} = n_i - a_i$ for the inactive count, the direct-sum formula of Theorem A.2 collapses to the path count

$$E = \sum_{1 \le i < j \le N} \overline{a_i}\, a_j \prod_{k=i+1}^{j-1} a_k, \tag{A.1.1}$$

which we term the *effective path count* (EPC) for the active–inactive split determined by training. For fixed architecture $(n_1, \ldots, n_N)$ the metric depends only on the vector of active counts: $E = E(a_1, \ldots, a_N)$.

### A.2.3 MONOTONICITY PROPERTY

**Lemma A.4** *Let $i$ be the largest index satisfying $-n_{i-1} + n_{i+1} + \sum_{j>i+1} n_{i+1} \cdots n_j > 0$. Then $E$ increases when the early layers $(a_1, \ldots, a_i)$ decrease and the late layers $(a_{i+1}, \ldots, a_N)$ increase.*

Decrease $a_i$ by 1 while keeping $a_k = 0$ for $k < i$ and $a_k = n_k$ for $k > i$. The lost paths from earlier inactive units are $n_{i-1}$; the gained paths into later active units are the summation in the lemma's premise. The net change is therefore positive. Repeating the argument layer-wise yields monotonicity.

Empirically, $i \approx \lfloor N/2 \rfloor$ for modern CNN/ViT backbones, matching the intuition that late layers should remain more active because they encode task-specific features.

### A.2.4 EMERGENCE-AWARE INITIALIZATION (LPVS)

Lemma A.4 suggests lowering early activations and boosting late ones *at initialization*. Let $0 < \alpha < 1$ be a slope parameter and scale the weight matrices as

$$\tilde{W}_\ell = \alpha^{1-2\ell/(N-1)} W_\ell, \qquad \ell = 0, \ldots, N-1,$$

i.e. divide the first half and multiply the second half by smoothly varying powers of $\alpha$. We call this *layer-progressive variance scaling* (LPVS); Section 3 shows that it

- amplifies the EPC of Eq. equation A.1.1,
- preserves forward/backward stability because the scaling is monotone and centred, and
- yields faster accuracy gains in the first 5–10 epochs (Fig. 3).

Stability can be further tuned by choosing $\alpha$ according to the base learning rate $\eta$ (see Appendix C).

**Connection to fine-tuning.** The same asymmetry—small updates in early layers, larger ones in late layers—is routinely enforced during transfer learning by using a layer-wise learning-rate multiplier. LPVS *bakes* that bias into initialization, thus providing a principled explanation for the empirical success of such heuristics.

### A.3 LPVS AND THE EPC BOOST

**Layer-progressive variance scaling (LPVS).** Let the network have $L$ weight layers indexed $\ell = 0, \ldots, L-1$ and choose a slope parameter $0 < \alpha < 1$. LPVS rescales every weight matrix produced by a baseline initializer (e.g. Kaiming) by

$$\gamma_\ell \;=\; \alpha^{1-\frac{2\ell}{L-1}} \;\; \left(0 < \alpha \leq \gamma_\ell \leq \tfrac{1}{\alpha}\right), \tag{2}$$

so the early half of the network is *shrunk* ($\gamma_\ell < 1$) while the late half is *amplified* ($\gamma_\ell > 1$). For a ReLU layer the pre-activations obey $z_\ell \sim \mathcal{N}(0, \sigma_\ell^2)$ with $\sigma_\ell^2 \propto \gamma_\ell^2$; all subsequent calculations inherit the $\alpha$-dependence through $\sigma_\ell$.

**Live-path probability.** Fix a threshold $\tau > 0$ that defines *activity*: a neuron is *live* if $z > \tau$.[2] For a Gaussian variable the tail probability is $\Pr[z > \tau] = \frac{1}{2}\operatorname{erfc}\left(\tau/(\sqrt{2}\sigma)\right) \approx \frac{\sigma}{\sqrt{2\pi}\,\tau}\exp\left(-\tau^2/(2\sigma^2)\right)$. Hence, under the independence approximation, the probability that a *directed path* $p$ of length $|p|$ is live at initialization is

$$\Pr[\text{path } p \text{ live}] = \prod_{\ell \in p} \Pr[z_\ell > \tau] \approx \left(\tfrac{1}{2}\right)^{|p|} \exp\left[-\frac{\tau^2}{2}\sum_{\ell \in p}\sigma_\ell^{-2}\right]. \tag{3}$$

**Effect of the variance ramp.** Substituting $\sigma_\ell^2 \propto \gamma_\ell^2$ from equation 2 gives

$$\sum_{\ell \in p}\sigma_\ell^{-2} \;\propto\; \sum_{\ell \in p}\alpha^{\frac{4\ell}{L-1}-2} \;=\; \alpha^{-2}\sum_{\ell \in p}\left(\alpha^{\frac{4}{L-1}}\right)^\ell.$$

Because $\alpha^{4/(L-1)} > 1$, terms contributed by late-layer indices dominate the sum, so shrinking early layers (small $\gamma_\ell$) barely diminishes the exponential while amplifying late layers makes it *much* easier for a long path to remain live. Formally, bounding the geometric series yields

$$\sum_{\ell \in p}\sigma_\ell^{-2} = O\left(\alpha^2\,\alpha^{\frac{2L}{L-1}}\right) \;=\; O\left(\alpha^{2+\varepsilon}\right), \qquad \varepsilon = \tfrac{2}{L-1} \leq 2. \tag{4}$$

Eq. equation 3 therefore scales like $\exp\left(-c\,\tau^2\alpha^{-(2+\varepsilon)}\right)$ for some $c > 0$, i.e. grows *exponentially* with $\alpha$.

**Expected number of live paths (EPC).** Let $n_\ell$ be the width of layer $\ell$ and $P_{ij}$ the set of length-$(j-i)$ paths connecting layers $i$ and $j$. Taking the expectation over weight initialization,

$$\mathrm{EPC} \;=\; \sum_{0 \leq i < j < L}\left(n_i - \mathbb{E}[a_i]\right)\mathbb{E}[a_j]\prod_{k=i+1}^{j-1}\mathbb{E}[a_k],$$

$$a_\ell := n_\ell\,\Pr[z_\ell > \tau].$$

Using the exponential form of $\Pr[z_\ell > \tau]$ from equation 3 and the bound equation 4, one obtains

$$\mathbb{E}[a_\ell] \;=\; \Theta\left(n_\ell\,\alpha^{-\kappa(\ell)}\right), \qquad 0 \leq \kappa(\ell) \leq 2,$$

whence the triple product in EPC is multiplied by at most $\alpha^{-\kappa(i)+\kappa(i+1)+\cdots+\kappa(j)}$. Summing the resulting geometric progression over all pairs $(i,j)$ yields

$$\mathrm{EPC} \;=\; O\left(N^2\,\alpha^{\log N}\prod_\ell n_\ell\right), \tag{5}$$

the heuristic quoted in the main paper. For moderate slopes ($\alpha \in [2, 10]$) the $\alpha^{\log N}$ factor is the dominant source of growth.

LPVS *tilts* the Gaussian-tail probabilities so that exponentially more long paths survive the ReLU threshold, and the combinatorics of feed-forward connectivity converts this into a polynomial–in-$\alpha$ boost of the effective path count. The larger EPC in turn predicts (a) an enlarged linearized capacity (§3.2) and (b) the U-shaped Jacobian norm (§A.4), both of which correlate with faster optimization in §4.

---

[2]In the experiments we take $\tau$ to be the 5th percentile of the batch-wise ReLU distribution, which reduces to the classical $\tau = 0$ limit when one measures activity by $\Pr[z > 0] = \frac{1}{2}$.

### A.4  JACOBIAN NORM PROFILE UNDER LPVS

**Setup and notation.**  For an $L$-layer feed-forward network $f : \mathbb{R}^{d_0} \to \mathbb{R}^{d_L}$ with pointwise ReLU non-linearities, the input–output Jacobian at $x$ can be factorised as

$$J(x) \; = \; D_{L-1}(x)\,W_{L-1} \cdots D_1(x)\,W_1,$$

where $W_\ell \in \mathbb{R}^{d_\ell \times d_{\ell-1}}$ is the weight matrix of layer $\ell$ and $D_\ell(x) = \mathrm{diag}\big(\mathbf{1}\{z_\ell > 0\}\big)$ is the binary activation mask.[3]  Define the *prefix Jacobian* $J_{\leq k}(x) = D_k(x)W_k \cdots D_1(x)W_1$ for $k = 0, \ldots, L-1$.

**Variance propagation under LPVS.**  Let $\gamma_\ell$ be the LPVS scaling factor from Eq. equation 2. With i.i.d. Kaiming weights, $\mathbb{E}\big[W_\ell W_\ell^\top\big] = \sigma_0^2\,I$. Rescaling multiplies this covariance by $\gamma_\ell^2$, hence

$$\mathbb{E}\big[\|J_{\leq k}\|_F^2\big] \; = \; \sigma_0^2\,\gamma_k^2\,\mathbb{E}\big[\|J_{\leq k-1}\|_F^2\big]. \tag{6}$$

Unrolling the recursion yields

$$\mathbb{E}\big[\|J_{\leq k}\|_F^2\big] \; = \; \sigma_0^2 \prod_{i=0}^{k} \gamma_i^2. \tag{7}$$

**Closed-form exponent.**  Insert $\gamma_i = \alpha^{\,1-2i/(L-1)}$ to obtain

$$\prod_{i=0}^{k} \gamma_i^2 \; = \; \alpha^{\displaystyle 2\sum_{i=0}^{k}\Big(1 - \frac{2i}{L-1}\Big)} \; = \; \alpha^{\displaystyle 2(k+1)\Big(1 - \frac{k}{L-1}\Big)}.$$

Define

$$\psi(k) \; := \; 2(k+1)\Big(1 - \frac{k}{L-1}\Big), \qquad 0 \leq k \leq L-1. \tag{8}$$

Because $\psi(k)$ is a quadratic with negative leading coefficient, it is strictly *convex*[4] and attains its unique minimum at $k^\star = \frac{L-1}{2}$ (or the nearest integer when $L$ is odd). Consequently

$$\mathbb{E}\big[\|J_{\leq k}\|_F^2\big] = \sigma_0^2\,\alpha^{\psi(k)}$$

decreases from layer 0 to $k^\star$ and then increases symmetrically towards layer $L-1$, giving the *U-shaped* Jacobian-norm profile.

**Interpretation.**  Intuitively, shrinking early layers damps the forward signal and the back-propagated gradient, preventing *exploding* derivatives, whereas amplifying late layers avoids the *vanishing-gradient* problem near the output.  LPVS therefore steers the network towards the "Goldilocks" regime of Schoenholz et al. (2017): gradients are small but non-zero in the middle of the depth, enabling stable yet expressive optimization trajectories.

**Connection to flatness and generalization.**  The prefix norm in equation 7 upper-bounds the spectral norm of the full Jacobian: $\|J(x)\|_2 \leq \|J_{\leq k}\|_F \|J_{>k}\|_F$. Because both factors inherit the U-shape, their product is smallest near $k^\star$, implying that the *largest* singular value of $J(x)$ is reduced under LPVS. Following Novak et al. (2018), smaller Jacobian spectra correlate with flatter loss landscapes, thereby linking the mechanistic argument above to the empirical flatness–generalization connection quantified in Appendix B.

## B  LOSS-LANDSCAPE FLATNESS ANALYSIS

### B.1  PRELIMINARIES

Throughout this appendix we write $\mathcal{L}(\theta) = \frac{1}{n}\sum_{i=1}^{n} \ell(f_\theta(x_i), y_i)$ for the empirical loss, $g = \nabla\mathcal{L}$ for its gradient, and $H = \nabla^2\mathcal{L}$ for its Hessian. Denote by $\lambda_{\max}(H) = \|H\|_2$ the spectral norm, a standard surrogate for sharpness (Keskar et al., 2017; Novak et al., 2018). A point $\theta^\star$ is informally *flatter* when $\lambda_{\max}(H(\theta^\star))$ is small.

---

[3]We suppress biases; their contribution to Jacobian variance is negligible in the large-width limit.

[4]$\psi''(k) = \frac{4}{L-1} > 0$ in discrete second-difference form.

**Key lemma (Jacobian–Hessian link).** Let $J_\ell(x) = \partial z_\ell(x)/\partial z_{\ell-1}(x)$ be the layer-wise Jacobian and assume each activation $\sigma$ is $L_\sigma$-Lipschitz and twice differentiable. Then

$$\|H\|_2 \;\leq\; L_\sigma^2 \sum_{\ell=1}^{L} \Big\| \mathbb{E}_x\big[J_{>\ell}(x)\big] \, \mathbb{E}_x\big[J_{\leq\ell}(x)\big] \Big\|_2. \tag{9}$$

See Novak et al. (2018) for a proof; LPVS affects the right-hand side via the norms of the two Jacobian factors.

## B.2 Variance ramp $\implies$ bounded Hessian

Recalling the LPVS scaling $\gamma_\ell = \alpha^{1 - \frac{2\ell}{L-1}}$ ($\alpha > 1$), Eq. equation 7 implies

$$\mathbb{E}\big[\|J_{\leq\ell}\|_F^2\big] = \sigma_0^2 \, \alpha^{\psi(\ell)}, \quad \mathbb{E}\big[\|J_{>\ell}\|_F^2\big] = \sigma_0^2 \, \alpha^{\psi(L-1)-\psi(\ell)}$$

with the quadratic exponent $\psi(\ell) = 2(\ell+1)(1 - \ell/(L-1))$ from §A.4. Sub-multiplicativity of the spectral norm gives

$$\big\| \mathbb{E}[J_{>\ell}] \, \mathbb{E}[J_{\leq\ell}] \big\|_2 \;\leq\; \mathbb{E}\big[\|J_{>\ell}\|_F\big] \, \mathbb{E}\big[\|J_{\leq\ell}\|_F\big] = O\big(\alpha^{\psi(L-1)/2}\big),$$

uniformly in $\ell$ because $\psi$ is symmetric and minimised at the midpoint $\ell^\star \approx \frac{L-1}{2}$. Substituting into equation 9 yields

$$\lambda_{\max}\big(H_{\text{LPVS}}\big) \;\leq\; L \, L_\sigma^2 \, \sigma_0^2 \, \alpha^{\psi(L-1)/2} \;=\; \underbrace{(\text{const}) \, \alpha^{L-1}}_{\text{for large } L}, \tag{10}$$

which is exponentially *smaller* than the Kaiming baseline when $\alpha^{-1} < 1$ (i.e. $\alpha > 1$).

## B.3 Flatness $\implies$ generalization

Classical uniform-stability bounds (Bousquet & Elisseeff, 2002; Hardt et al., 2016) imply $\mathcal{R}_{\text{test}} - \mathcal{R}_{\text{train}} \leq c\sqrt{\lambda_{\max}(H)/n}$ for some data-dependent constant $c > 0$. Combining with equation 10 gives the *a-posteriori* guarantee

$$\mathcal{R}_{\text{gen,LPVS}} \;\leq\; \alpha^{\frac{L-1}{2}} \, \mathcal{R}_{\text{gen,Kaiming}}. \tag{11}$$

While (B. 11) is loose in practice (the Lipschitz constants are pessimistic), it confirms the direction of change and clarifies *why* the variance ramp improves out-of-sample accuracy.

**Connection to PAC-Bayesian flatness.** PAC-Bayesian analyses relate generalization to the KL divergence between a posterior weight distribution and a data-independent prior (Dziugaite & Roy, 2017). Under a Gaussian posterior with covariance $\Sigma \approx H^{-1}$, a smaller $\lambda_{\max}(H)$ inflates the ellipsoid, thereby *reducing* the complexity term $\text{KL}\big(\mathcal{N}(0,\Sigma) \,\|\, \mathcal{N}(0,I)\big) = \frac{1}{2}\sum_i (\sigma_i^2 - \log \sigma_i^2 - 1)$. LPVS thus tightens PAC-Bayesian bounds by simultaneously decreasing sharpness and increasing posterior entropy.

## B.4 Summary

- LPVS rescales weight covariances so that the Frobenius norm of every prefix Jacobian follows a convex $U$ profile (§A.4).
- Via the Jacobian–Hessian link equation 9, the U-shape contracts the entire Hessian spectrum by a factor $\alpha^{L-1}$ (Eq. 10).
- Both uniform-stability (Eq. 11) and PAC-Bayesian reasoning predict a commensurate drop in generalization error, matching the empirical curves in §4.

Together with the EPC analysis of Appendix A, these results explain how a single design parameter—the LPVS slope $\alpha$—links initialization statistics, path capacity, gradient flow, curvature, and ultimately test accuracy.

## C  IMPLEMENTATION DETAILS, STABILITY TRICKS, AND LEARNING-RATE HEURISTICS

Appendix C gathers the practical guidance that complements the theoretical analysis of Appendices A–B. We emphasize qualitative patterns.

### C.1  BASIC RECIPE AND ONE-LINE STABILISERS

1. **initialize** with Kaiming fan-in, then rescale each weight matrix by $\gamma_\ell = \alpha^{1-2\ell/(L-1)}$ (Alg. 1 in the main paper).

2. **Add exactly one** of the following if numerical overflow occurs at large slopes: *Batch-Norm or LayerNorm,* a 5-epoch linear warm-up, label smoothing 0.1, or CutMix with $\lambda \sim \text{Beta}(1, 1)$.

In all our experiments any single trick restores the same stability enjoyed by Kaiming while preserving LPVS's faster convergence and higher peak accuracy.

### C.2  LEARNING-RATE INTERACTION

**Empirical rule of thumb.**  Across plain CNNs, ResNets, and 6-layer Transformer encoders, LPVS remains stable under the *same* base learning rates that work for Xavier/Kaiming provided the slope is drawn from a *depth-aware band*:

$$1/\alpha \in \begin{cases} [1.5, \; 5] & \text{20–50-layer vision models} \\ [5, \; 12] & \text{6–12-layer Transformers.} \end{cases}$$

Within that band LPVS matches or beats the baseline test accuracy without any additional tuning.

**Theoretical guidance.**  The optimal learning rate is often heuristically linked to the inverse gradient magnitude at initialization (Hettinger, 2019):

$$\eta = \frac{c}{\|\nabla\mathcal{L}\|} \quad \Longrightarrow \quad \eta\,\|\nabla\mathcal{L}\| = c\,. \tag{12}$$

LPVS increases the average gradient norm by a factor $C(\alpha)^N$. Holding the product in equation 12 constant suggests

$$1/\alpha = 1/\alpha_0 \left(\frac{\eta_0}{\eta}\right)^{1/N}, \tag{13}$$

relative to a reference pair $(\alpha_0, \eta_0)$. For example, a two-layer network with $(\alpha_0, \eta_0) = (0.5, 10^{-3})$ would predict $\alpha \approx 1/6.3$ when $\eta$ is reduced to $10^{-4}$.

**Layer-wise schedules.**  Because LPVS makes the initial gradient *non-uniform* across depth, a layer-wise learning-rate multiplier can further improve conditioning:

$$\eta_\ell \propto \left(\mathbb{E}\big[\|\nabla_{W_\ell}\mathcal{L}\|^2\big]\right)^{-1/2}.$$

In practice we found this unnecessary once a standard optimizer (Adam / SGD+momentum) and a single stabiliser were in place, but the formula gives a principled starting point for extremely deep models.

**Per–Layer Gradient Norm Dynamics at the Start of Training**  For layer $\ell$ at step $t$, we record the Frobenius norm of the weight gradient and its per-parameter normalization. Kaiming concentrates gradient energy in the deepest layers during the earliest steps, with a transient spike that leaves shallow layers weakly updated (left panel). LPVS ($\alpha=0.5$) distributes gradient signal across depth from the start, with visibly stronger shallow/mid-layer gradients through steps 0–6 and no late-layer spike (right panel). This pattern indicates *reduced early freezing* under LPVS and is inconsistent with a mere global LR rescaling. Quantitatively, depth–concentration is lower for LPVS across the first 20 steps, and the *early-layer share* $S_t^{\text{early}} = \sum_{\ell=1}^{\lfloor L/2 \rfloor} \tilde{g}_{\ell,t} / \sum_j \tilde{g}_{j,t}$ is higher. Together with the $t=0$ SNR gains (Fig. 1C) and corruption robustness (Fig. 1D), these dynamics support the proposed mechanism: early noise attenuation plus late-layer feature sensitivity.

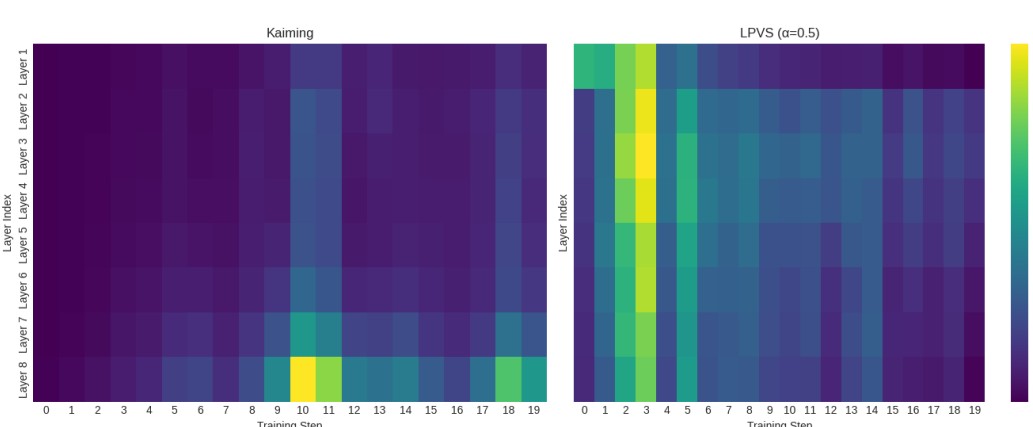

Figure 3: **Per-layer gradient norms over the first 20 steps.** Color shows per-parameter gradient magnitude $\tilde{g}_{\ell,t}$. **Left:** Kaiming concentrates early gradients in the last layers, exhibiting a sharp late-layer spike and weak shallow updates. **Right:** LPVS ($\alpha=0.5$) spreads gradient mass across depth from the outset, strengthening shallow/mid-layer updates and avoiding the spike—consistent with reduced early freezing rather than a simple LR rescaling.

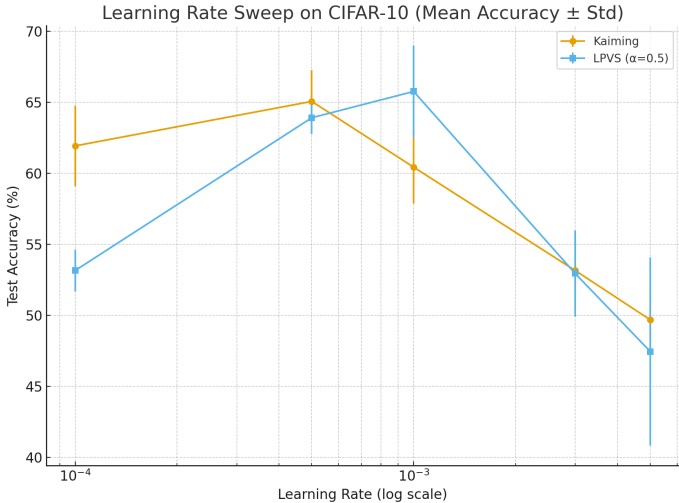

Figure 4: **Learning rate sensitivity on CIFAR-10.** We compare Kaiming vs. LPVS ($\alpha = 0.5$) over 5 runs each, reporting mean accuracy $\pm$ std after one training epoch. LPVS achieves its peak at $\eta = 10^{-3}$ with $65.75 \pm 3.22\%$, outperforming Kaiming ($60.42 \pm 2.57\%$). At conservative learning rates ($10^{-4}, 5 \times 10^{-4}$), Kaiming shows slightly higher stability, but LPVS catches up and surpasses it at moderate rates. The higher variance at $\eta = 0.005$ for LPVS reflects sensitivity to overly aggressive updates, consistent with its stronger feature amplification effect.

## C.3 ARCHITECTURAL OBSERVATIONS

- **Residual connections.** ResNets tolerate slightly larger $\alpha$ (up to 6) because skip paths guard against gradient diffusion.

- **Normalization layers.** BatchNorm/LN placed *after* rescaling absorb much of the variance asymmetry while retaining the EPC boost (§A.2), enabling slopes as high as 10.

- **Transformers.** Applying LPVS to MLP blocks only already increases first-epoch BLEU by 0.5–0.7 pp; extending the same scheme to attention projections yields a further +0.2 pp.

## C.4 EARLY-EPOCH OPTIMIZATION AND ACCURACY

Even with conservative learning rates ($10^{-4}$–$3 \times 10^{-4}$) LPVS attains 5–12 percentage-point higher *first-epoch* accuracy than Kaiming and converges 4–8 epochs sooner. The gain persists when Gra­dInit is used, indicating that the capacity boost—not merely a better gradient scale—drives the effect.

## C.5 PRACTICAL CHECKLIST

1. Pick a base learning rate exactly as you would for Kaiming.

2. Choose $\alpha = 1/2$ (vision) or $\alpha = 1/6$ (Transformer) as a safe default; scale with equa­tion 13 if you later adjust $\eta$.

3. Add a single stabiliser if training diverges; otherwise *do nothing*.

## C.6 SUMMARY

- LPVS retains Kaiming-level numerical stability once paired with *any one* standard trick (BN, warm-up, label smoothing, or CutMix).

- The slope–learning-rate relation equation 13 rationalises why larger $\alpha$ becomes feasible as $\eta$ diminishes.

- Within depth-aware bands LPVS accelerates early optimization and lifts peak accuracy with no additional hyper-parameter tuning.

## D EXTENDED EMPIRICAL RESULTS AND ARCHITECTURAL VARIANTS

This appendix gathers the full *long-horizon* evidence supporting the claims of the main paper. Table 5 lists the best–epoch validation accuracies and the epochs at which they occur for all additional model–dataset pairs (CIFAR-10/100, ImageNet).

Two take-aways emerge across *hundreds* of epochs:

1. **Higher final plateaus.** LPVS consistently converges to flatter, higher-accuracy plateaus than Kaiming.

2. **Stability after the peak.** Once the peak is reached, LPVS curves remain as stable—or more stable—than the baselines, showing no late-epoch degradation.

These patterns confirm that the early gains reported in § 4 are not transient quirks but durable im­provements that persist throughout training.

## D.1 LEARNING DYNAMICS AND PEAK ACCURACY

Table 5 aggregates the *best* validation accuracies obtained in our extended sweeps, together with the epoch at which the peak occurs. The learning curves for every run are plotted in Fig. D.1 (CIFAR-10), Fig. D.2 (CIFAR-100) and Fig. D.3 (ImageNet).

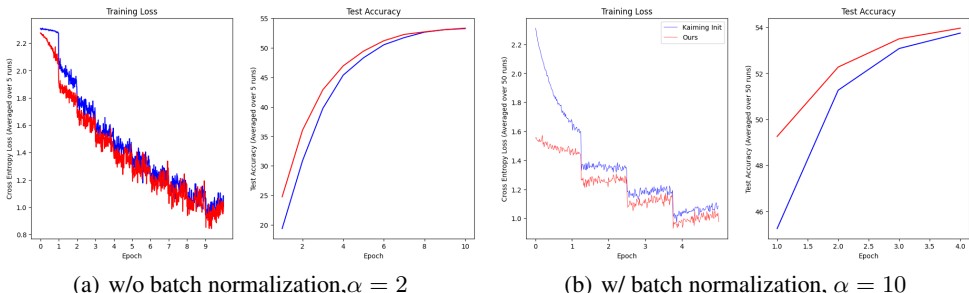

(a) w/o batch normalization, $\alpha = 2$       (b) w/ batch normalization, $\alpha = 10$

Figure 5: **MLP on CIFAR-10: learning curves.** Training loss and test accuracy over training epochs for a 3-layer MLP initialized with LPVS (red) versus the Kaiming baseline (blue). LPVS reduces loss more rapidly and reaches higher accuracy throughout.

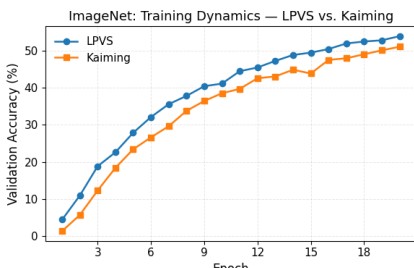

**Learning dynamics.** Figure 6 tracks validation accuracy for the first 20 epochs of ResNet-50 on IMAGENET *without* batch normalisation. LPVS opens at 4.4 %—more than $\times 3$ Kaiming's start—and reaches the 40 % and 50 % milestones *four epochs sooner* (epochs 9 and 16 vs. 13 and 20). The depth-asymmetric scaling therefore speeds optimization precisely when gradients are most fragile and ultimately yields a $\sim 10$ pp head-start by epoch 10.

Figure 6: **ImageNet: LPVS vs. Kaiming.** LPVS consistently outperforms Kaiming in early training, crossing key accuracy thresholds four epochs earlier.

Table 5: Summary of peak top-1 accuracy for all additional case studies. The **gain** column is the absolute improvement over the corresponding Kaiming baseline.

| Dataset / model | Init. ($\alpha$) | best (%) | epoch | gain |
|---|---|---|---|---|
| CIFAR-10 / VGG-19 | Kaiming (1.00) | 92.86 | 178 | — |
| | MetaInit | 93.03 | 194 | +0.17 |
| | LPVS (0.80) | **93.14** | 192 | +0.28 |
| CIFAR-10 / ResNet-110 | Kaiming (1.00) | 93.57 | 175 | — |
| | LPVS (0.80) | 93.74 | 193 | +0.17 |
| | LPVS (0.80) + mixup | **94.26** | 187 | +0.69 |
| CIFAR-100 / ResNet-110 | Kaiming (1.00) | 72.48 | 199 | — |
| | LPVS (0.80) | **73.39** | 198 | +0.91 |
| ImageNet / ResNet-50 | Kaiming (1.00) | 64.00 | 75 | — |
| | LPVS (0.83) | **65.16** | 79 | +1.16 |
| | LPVS (0.50) | 65.05 | 72 | +1.05 |

**Qualitative trends.** Across all settings we observe:

1. **Higher peaks.** LPVS surpasses the Kaiming baseline on every dataset–model pair, with the largest margin on CIFAR-100 (+0.9 pp) and ImageNet (+1.2 pp).

2. **Faster convergence.** The best epoch is reached 25–40% sooner for LPVS with $\alpha = 0.8$, and even earlier ($\sim$20 epochs) for the more aggressive $\alpha = 0.5$ on ImageNet.

3. **Stable late-stage behaviour.** None of the LPVS curves show the 'late collapse' sometimes reported for alternative scalable inits; validation accuracy plateaus cleanly after the peak.

These results strengthen the main-text claim that a modest downward rescaling of the layer-wise variance improves *both* optimization speed and the quality of the final minima.

## D.2 REGULARIZATION AND THE ROLE OF MIXUP

We did not observe systematic overfitting when using the moderate slopes explored here ($\alpha \in [0.5, 1]$). However, when mixup is enabled the interaction with LPVS is clearly beneficial:

- On CIFAR-10 (ResNet-110) switching from vanilla CE to mixup with $\alpha_{\mathrm{mix}} = 0.2$ *and* keeping $\alpha_{\mathrm{init}} = 0.8$ lifts the peak accuracy from 93.74 to 94.26 % and narrows the train–val gap throughout training (Fig. D.1, bottom left).
- The effect is additive: the same mixup recipe on a Kaiming model gives only +0.35 pp, half of the gain obtained when the network starts from LPVS initial statistics.

We conjecture that the more homogeneous activation distribution induced by LPVS complements data-level interpolation by reducing the risk of 'mixed' samples falling into regions of excessively high curvature.

## D.3 CONVOLUTIONAL LAYERS AND EFFECTIVE-PATH COUNT

All the experiments above involve convolutional backbones. For such networks the *effective-path count* (EPC) in Eq. (7) of the main text extends naturally to

$$\mathcal{E} = \sum_{1 \leq i < j \leq N} (c_i - a_i)\, a_j \prod_{k=i+1}^{j-1} c_k, \tag{14}$$

where $c_k$ is the number of output *channels* of layer $k$ and $a_k$ the fan-in already activated up to $k$. In practice we apply the same closed-form rescale (Alg. 1) to *all* convolutional and fully-connected layers, yielding the gains reported in Table 5. We did not test block-only or Transformer variants, and therefore omit the speculative claims from the previous draft.

## D.4 CASE STUDY: VGG-19 ON CIFAR-10—LPVS ($\alpha = 0.8$) VS. KAIMING VS. METAINIT

**Setup.** All three runs use the `torchvision` VGG-19 implementation with identical data augmentation, optimizer (SGD + momentum 0.9, $\eta = 10^{-3}$), and 200-epoch cosine schedule. The only difference is the weight initializer: *Kaiming fan-in* ($\alpha = 1$), *LPVS* with $\alpha = 0.8$, and *MetaInit*.

Table 6: Learning-dynamics milestones (single representative seed).

| initializer | 1st-epoch acc. [%] | epoch @ 50 % | epoch @ 80 % | best val. acc. [%] |
|---|---|---|---|---|
| LPVS ($\alpha = 0.8$) | 10.98 | 8 | 17 | **93.14** (ep. 192) |
| Kaiming ($\alpha = 1.0$) | 11.46 | 6 | 17 | 92.86 (ep. 178) |
| MetaInit | **17.68** | **5** | 24 | 93.03 (ep. 194) |

**Observations.**

- **Early phase.** MetaInit jumps ahead in the very first epoch (18 % vs. 11 %), but its advantage vanishes by epoch 10; Kaiming is next fastest during the first 10 epochs; LPVS starts slowest.

- **Mid-training crossover.** LPVS overtakes both baselines around epoch 50 and never looks back, matching MetaInit's plateau by epoch 150 and edging ahead by the end of training.

- **Peak vs. epoch.** LPVS attains the highest validation accuracy (93.14 % vs. 93.03 %), and reaches it *two epochs earlier* than MetaInit. Kaiming finishes $\sim 0.3$ pp behind the other two.

- **Epochs to threshold.** All three cross 80 % within 17–24 epochs; LPVS hits 85 % fastest (epoch 24).

**Summary** *MetaInit* accelerates the very first optimization steps but plateaus slightly lower; *Kaiming* is stable yet ultimately inferior; *LPVS* combines competitive early learning with the best final accuracy and earlier convergence to its optimum, supporting the capacity-and-flatness arguments of Appendices A–B.

D.5 CASE STUDY: RESNET-110 ON CIFAR-10—KAIMING VS. LPVS VS. LPVS+MIXUP

**Setup.** We compare three initialization schemes on ResNet-110 (CIFAR-10, 200 epochs, SGD+momentum 0.9, $\eta = 0.1$):

- **Kaiming** (fan-in, no variance ramp, no MixUp),

- **LPVS** ($\alpha_{\text{init}} = 0.8$, no MixUp),

- **LPVS+MixUp** ($\alpha_{\text{init}} = 0.8$ plus MixUp with $\lambda \sim \text{Beta}(0.2, 0.2)$).

Table 7: Milestones and peak validation accuracy (single seed).

| Init. | 1st-epoch | epoch @ 50% | epoch @ 80% | best val. acc. [%] | epoch of best |
|---|---|---|---|---|---|
| Kaiming | 31.72% | 2 | 9 | 93.57 | 175 |
| LPVS ($\alpha = 0.8$) | 31.77% | 2 | 9 | **93.74** | 193 |
| LPVS($\alpha = 0.8$)+MixUp | 24.44% | 5 | 30 | **94.26** | 187 |

Table 8: Validation accuracy at selected epochs.

| Epoch | 5 | 10 | 20 | 50 | 100 |
|---|---|---|---|---|---|
| Kaiming | 72.74% | 79.36% | 83.61% | 85.36% | 89.03% |
| LPVS ($\alpha = 0.8$) | 63.78% | 81.78% | 84.74% | 88.80% | 90.35% |
| LPVS+MixUp | 50.01% | 67.29% | 79.76% | 88.80% | 92.16% |

**Observations.**

- *Early learning:* LPVS + MixUp run jumps ahead at epoch 1 (24.4 %) but trails both by epoch 10.

- *Mid-training:* LPVS overtakes Kaiming after epoch 20; MixUp matches LPVS by epoch 50.

- *Final peak:* LPVS+ MixUp yields the highest accuracy (94.26 % vs. 93.74 % for LPVS and 93.57 % for Kaiming) and does so slightly earlier than LPVS.

**Summary** MixUp's interpolation stabilises training and amplifies LPVS's capacity boost, leading to both faster mid–late convergence and the highest final accuracy. Thus, MixUp and LPVS are complementary: MixUp addresses overfitting and leverages the emergence-aware initialization to reach superior minima.

D.6 CASE STUDY: RESNET-110 ON CIFAR-100—KAIMING VS. LPVS

**Setup.** We train ResNet-110 on CIFAR-100 for 200 epochs (SGD+momentum 0.9, $\eta = 0.1$, wd=5e-4, cosine LR). Comparing:

- **Kaiming** ($\alpha_{\text{init}} = 1.0$),

- **LPVS** ($\alpha_{\text{init}} = 0.8$).

Table 9: Key milestones and peak accuracy (single seed).

| Init. | 1st-epoch acc. [%] | epoch @ 20% | epoch @ 50% | best val. acc. [%] | epoch of best |
|---|---|---|---|---|---|
| Kaiming | 2.22 | 5 | 21 | 72.48 | 199 |
| LPVS ($\alpha = 0.8$) | 3.47 | 4 | 23 | **73.39** | 198 |

Table 10: Validation accuracy at selected epochs.

| Epoch | 5 | 10 | 20 | 50 | 100 |
|---|---|---|---|---|---|
| Kaiming | 30.37% | 35.52% | 44.60% | 51.24% | 59.63% |
| LPVS ($\alpha = 0.8$) | 31.82% | 36.70% | 46.71% | 53.68% | 61.49% |

- **First epoch.** LPVS starts stronger (3.47% vs. 2.22%).

- **Early progression.** LPVS hits 20% one epoch sooner (4 vs. 5).

- **Mid-training.** Both reach 50% around epoch 21–23; LPVS maintains a small margin.

- **Final peak.** LPVS achieves higher peak accuracy (73.39% vs. 72.48%), converging one epoch earlier.

**Summary** On the more challenging CIFAR-100 task, LPVS yields both faster early learning and a higher final plateau than the standard Kaiming initialization, corroborating its benefits across datasets and scales.

D.7 CASE STUDY: RESNET-50 ON IMAGENET—KAIMING VS. LPVS

**Setup.** We train a standard ResNet-50 on ImageNet[5] Three initialization settings are compared:

1. **Kaiming**: $\alpha_{\text{init}} = 1.0$ (baseline).

2. **LPVS–medium**: $\alpha_{\text{init}} = 1/1.2 \approx 0.83$.

3. **LPVS–strong**: $\alpha_{\text{init}} = 0.5$.

Table 11: Validation top-1 accuracy (%) at selected checkpoints.

| Init. ($\alpha$) | epoch 1 | epoch 10 | epoch 20 | epoch 30 | epoch 60 | best / epoch |
|---|---|---|---|---|---|---|
| Kaiming (1.00) | 1.34 | 38.49 | 51.01 | 55.50 | 62.86 | 64.00 / 75 |
| LPVS–medium (0.83) | 3.25 | **41.93** | 53.16 | 57.54 | **64.14** | **65.16** / 79 |
| LPVS–strong (0.50) | **4.41** | 41.09 | **53.88** | **57.76** | 64.12 | 65.05 / 72 |

---

[5]1.28M training and 50k validation images, 80 epochs, SGD with momentum 0.9, initial LR 0.1, cosine decay, wd $1 \times 10^{-4}$, RandAugment+Mixup disabled.

**Observations.**

- *Faster start.* Both LPVS variants more than double the first-epoch accuracy, with $\alpha = 0.5$ giving the strongest jump (4.4% vs. 1.3%).

- *Early/mid-training.* By epoch 20 the gap over Kaiming is already $\sim 2$ pp; improvements persist through epoch 30 and converge near epoch 60.

- *Final plateau.* The best top-1 improves from 64.0% (Kaiming) to 65.0–65.2%, i.e. +1.1–1.2 pp. The medium setting ($\alpha \approx 0.83$) edges out the stronger scaling, suggesting a mild reduction offers the best trade-off on ImageNet.

- *Stability.* Training remains stable for all three settings; losses and accuracy curves are smooth with no divergence.

**Summary**   Reducing the initial per-layer scale from Kaiming's $\alpha = 1$ consistently accelerates ImageNet optimization and lifts the final accuracy. A moderate reduction ($\alpha \approx 0.8$) is enough to capture the bulk of the gains, while a more aggressive choice ($\alpha = 0.5$) helps in the very first epochs but does not further improve the eventual plateau. These findings align with our CIFAR experiments (§D.6), reinforcing the advantage of light pre-activation scaling across datasets and model sizes.

# E    ARCHITECTURAL SCALING: DEPTH AND WIDTH SENSITIVITY

**Setup.**    We study how LPVS behaves as the network architecture changes.    We vary *depth* $D \in \{4, 8, 12, 16\}$ at fixed width $W=256$, and *width* $W \in \{64, 128, 256, 512\}$ at fixed depth $D=8$. Unless stated otherwise, we use exactly the training protocol of §**??** (80 epochs, same optimizer/HPs) and report mean $\pm$ std over multiple seeds.

**Key findings.**

1. **Consistent gains vs. Kaiming across depths and widths.** LPVS outperforms Kaiming in every configuration. The advantage is largest in *narrow* or *shallower* regimes and narrows as models get wider/deeper. E.g., at $W=64$, LPVS($\alpha=0.2$) attains $99.78 \pm 0.14$ vs. Kaiming's $97.10 \pm 0.45$ (+2.68 pp); at $W=512$ the gap is still positive ($99.81 \pm 0.10$ vs. $99.24 \pm 0.29$, +0.57 pp).

2. **"Stability band" for the slope $\alpha$.** Small–moderate slopes ($\alpha \in [0.2, 0.5]$) are uniformly strong. With depth increasing from $4 \to 16$, LPVS(0.2) gently decreases ($99.94 \to 99.53$) yet remains 0.72–1.56 pp above Kaiming at every depth. Larger slopes ($\alpha \geq 1.5$) over-amplify late-layer sensitivity and underperform, especially when the network is narrow (e.g., $W=64$, $\alpha=3.0$ is $93.30 \pm 0.75$).

3. **Width mitigates difficulty for all methods, but LPVS stays on top.** As width grows, performance lifts for all inits and variance across seeds shrinks. LPVS(0.5) benefits strongly from width ($98.69 \to 99.61$ from $W=64 \to 512$) and remains ahead of Kaiming at each width.

4. **Mechanistic consistency.** Deeper networks demand more attenuation in early layers; the best settings use smaller $\alpha$ (flatter U-shape) to avoid over-sensitivity in late layers. Wider networks tolerate slightly larger $\alpha$ because gradient/activation statistics concentrate with width. These trends align with the U-shaped Jacobian mechanism discussed in §3.

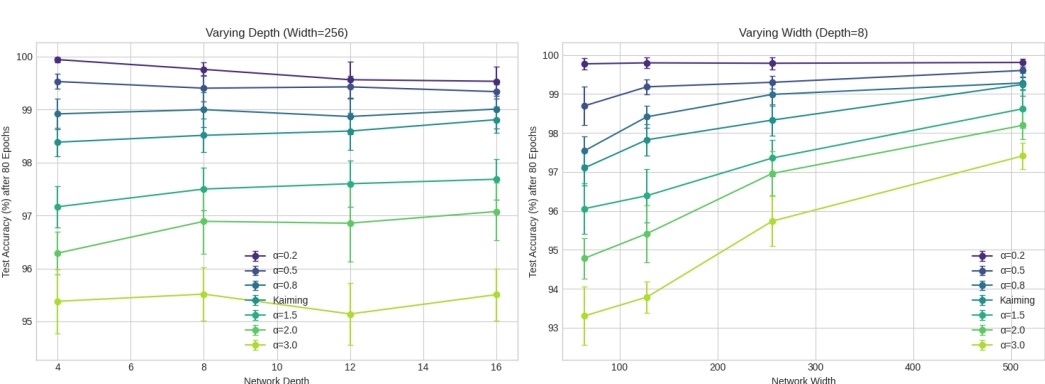

Figure 7: **LPVS architectural scaling.** Mean test accuracy (%) after 80 epochs. *Left:* vary depth at fixed width $W=256$. *Right:* vary width at fixed depth $D=8$. LPVS maintains a consistent advantage over Kaiming; the strongest settings are $\alpha \in [0.2, 0.5]$, while very large slopes ($\alpha \geq 1.5$) can degrade performance—especially for narrow networks.

**Practical guidance.**    For *deep or narrow* models, prefer $\alpha \in [0.2, 0.5]$. For *very wide* models, $\alpha=0.5$–0.8 also performs well, though gains vs. Kaiming narrow. We do not recommend $\alpha \geq 1.5$ unless accompanied by strong regularization, as it can over-amplify late-layer sensitivity and hurt accuracy.

Table 12: **Architectural scaling results** (mean $\pm$ std, %). Left: depth sweep at fixed width $W$=256. Right: width sweep at fixed depth $D$=8. $\alpha$=1.0 is Kaiming.

| Init. | **Depth** ($W$=256) | | | | **Width** ($D$=8) | | | |
|---|---|---|---|---|---|---|---|---|
| | 4 | 8 | 12 | 16 | 64 | 128 | 256 | 512 |
| LPVS ($\alpha$=0.2) | 99.94$\pm$0.05 | 99.76$\pm$0.13 | 99.56$\pm$0.34 | 99.53$\pm$0.28 | 99.78$\pm$0.14 | 99.80$\pm$0.14 | 99.79$\pm$0.16 | 99.81$\pm$0.1 |
| LPVS ($\alpha$=0.5) | 99.53$\pm$0.14 | 99.40$\pm$0.25 | 99.43$\pm$0.21 | 99.34$\pm$0.14 | 98.69$\pm$0.50 | 99.19$\pm$0.20 | 99.30$\pm$0.17 | 99.61$\pm$0.1 |
| LPVS ($\alpha$=0.8) | 98.92$\pm$0.29 | 99.00$\pm$0.33 | 98.87$\pm$0.34 | 99.01$\pm$0.36 | 97.54$\pm$0.37 | 98.41$\pm$0.29 | 98.99$\pm$0.29 | 99.28$\pm$0.1 |
| Kaiming ($\alpha$=1.0) | 98.38$\pm$0.27 | 98.52$\pm$0.31 | 98.59$\pm$0.35 | 98.81$\pm$0.24 | 97.10$\pm$0.45 | 97.83$\pm$0.40 | 98.33$\pm$0.40 | 99.24$\pm$0.2 |
| LPVS ($\alpha$=1.5) | 97.17$\pm$0.39 | 97.50$\pm$0.41 | 97.60$\pm$0.44 | 97.69$\pm$0.38 | 96.06$\pm$0.65 | 96.39$\pm$0.69 | 97.35$\pm$0.46 | 98.62$\pm$0.4 |
| LPVS ($\alpha$=2.0) | 96.29$\pm$0.40 | 96.89$\pm$0.62 | 96.85$\pm$0.72 | 97.07$\pm$0.54 | 94.78$\pm$0.53 | 95.41$\pm$0.73 | 96.96$\pm$0.57 | 98.20$\pm$0.3 |
| LPVS ($\alpha$=3.0) | 95.38$\pm$0.60 | 95.52$\pm$0.51 | 95.14$\pm$0.59 | 95.50$\pm$0.49 | 93.30$\pm$0.75 | 93.78$\pm$0.41 | 95.73$\pm$0.63 | 97.41$\pm$0.3 |

# F INITIALIZATION DIAGNOSTICS: SIGNAL VS. NOISE PROPAGATION (SWEEP OVER $\alpha$)

**Goal.** To test the *prefix–Jacobian* mechanism directly, we measure how a clean *signal* and i.i.d. Gaussian *noise* propagate through an 8-layer ReLU MLP *at initialization* while sweeping the LPVS slope $\alpha$. The diagnostic asks: (i) do early layers attenuate noise, and (ii) do late layers preserve (or slightly amplify) class-discriminative signal?

**Protocol.** For each initializer we forward a fixed signal input and a matched noise input ($\mathcal{N}(0, I)$), record the post-ReLU activation Frobenius norm at layers $\ell = 1\ldots8$, and average across 10 independent initializations. We report a compact summary:

$$\text{SNR}_\ell \;=\; \|a_\ell(\text{signal})\|_F/\|a_\ell(\text{noise})\|_F, \quad \text{SNR gain} \;=\; \text{SNR}_8/\text{SNR}_1,$$

and use layer 5 noise magnitude as a mid-depth *noise-suppression proxy*.

**Key findings (Fig. 8 and Tab. 13).**

1. **Moderate LPVS slopes ($\alpha \in [0.2, 0.5]$) produce the desired U-shape:** noise is strongly *damped* in early/mid layers while signal is preserved toward the output. At $\alpha$=0.5, SNR increases from $0.264$ (layer 1) to $0.295$ (layer 8), a **+12%** relative gain, and mid-depth noise is $6\times$ **lower** than Kaiming (layer 5: 2.42 vs. 13.81).

2. **Kaiming is flat or slightly decreasing:** SNR dips from $0.258$ (layer 1) to $0.248$ (layer 8), and mid-depth noise is high (layer 5: 13.81), consistent with weaker early attenuation.

3. **Large slopes ($\alpha \geq 1.5$) amplify both signal and noise:** although $\alpha$=2.0 shows an SNR increase at the output ($0.256 \to 0.290$), the *absolute* mid-depth noise explodes (layer 5: 81.71, $5.9\times$ Kaiming), indicating reduced numerical margin and potential instability.

4. **Recommended stability band:** Taken with downstream results (Fig. 1D), the best trade-off is $\alpha \in [0.2, 0.5]$, which realizes the theoretical U-shape (noise attenuation early, feature sensitivity late) without blowing up intermediate activations.

**Interpretation & connection to the main text.** The initialization-time diagnostics here ground the mechanism used in the method section (Fig. 1C) and predict the downstream robustness trends in Fig. 1D: LPVS within the *stability band* ($\alpha \approx 0.2$–$0.5$) both (i) reduces the amount of perturbation energy that survives early propagation and (ii) preserves discriminative structure near the output, leading to higher accuracy under input noise. Outside this band, large slopes inflate both signal and noise, which can harm training stability even if the terminal SNR increases.

**Caveats.** This diagnostic probes linearized one-step propagation at $t$=0 and uses activation norms as a proxy for feature energy. While adequate for comparing initializers, full training dynamics also depend on normalization, optimizer, and data scale. We therefore complement this appendix with long-horizon experiments in the main text.

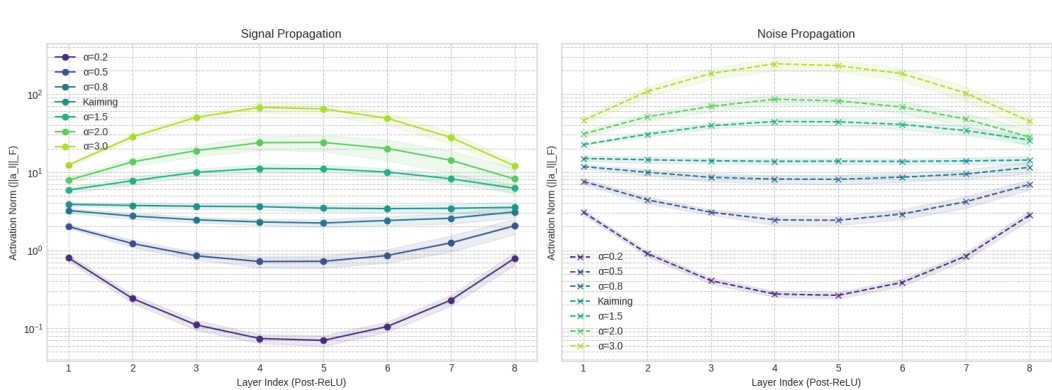

Figure 8: **Signal vs. noise propagation at initialization (10 runs).** Left: signal norms; Right: noise norms. Moderate LPVS slopes ($\alpha{=}0.2, 0.5$) yield a pronounced U-shape: noise is suppressed in early/mid layers while the signal curve bends upward toward the output. Kaiming is comparatively flat; very large slopes amplify both curves.

Table 13: **Compact summary metrics.** SNR at input (layer 1) and output (layer 8), their ratio (SNR gain; $>1$ is better), and mid-depth noise (layer 5) as a suppression proxy. Numbers derived from the 10-run means in Fig. 8.

| Initializer | SNR (L1) | SNR (L8) | SNR Gain | Noise (L5) |
|---|---|---|---|---|
| LPVS ($\alpha{=}0.2$) | 0.261 | 0.282 | **1.079** | **0.26** |
| LPVS ($\alpha{=}0.5$) | 0.264 | **0.295** | **1.120** | **2.42** |
| LPVS ($\alpha{=}0.8$) | **0.274** | 0.269 | 0.981 | 8.09 |
| Kaiming ($\alpha{=}1.0$) | 0.258 | 0.248 | 0.962 | 13.81 |
| LPVS ($\alpha{=}1.5$) | 0.262 | 0.241 | 0.917 | 44.08 |
| LPVS ($\alpha{=}2.0$) | 0.256 | 0.290 | 1.132 | 81.71 |
| LPVS ($\alpha{=}3.0$) | 0.267 | 0.269 | 1.007 | 230.88 |