# OpenReview forum: "Layer-Scaled Weight Initialization for Efficient Deep Neural Network Optimization"
_ICLR.cc/2026/Conference — Submitted to ICLR 2026_

### Official Review · Reviewer_AJWU · 2025-10-31

**Soundness:** 2
**Presentation:** 1
**Contribution:** 3
**Rating:** 4
**Confidence:** 3

**Summary:**

The authors propose a novel method for weight initialization called LPVS. Specifically, they suggest rescaling existing standard initializations based on network depth, using smaller values for shallow layers and larger values for deep layers. The authors claim that this prevents dead neurons in shallow layers and improves signal propagation in deeper layers.

**Strengths:**

- The authors suggest a novel weight initialization that focuses on preventing dead ReLUs in shallow layers and improving signal propagation in deeper layers.
- Their method can be seamlessly integrated with current practices.
- They experimentally demonstrate the benefit of their initialization method.

**Weaknesses:**

- Regarding the concept of Effective Path Count, the main body does not clearly distinguish the authors' novel contributions from existing work.
- In line 233, it is mentioned that there might be stability issues that can be addressed with standard methods, such as warm-up or normalization layers. Although this is discussed again in the Appendix, a more in-depth discussion with experimental support on this topic would be important, as it is crucial in practice.
- The authors mention "pixel-level perturbation" but do not provide sufficient context for what they mean by this term.
- In the experimental section lacks important technical details, such as which standard initialization is used as the base for the LPVS wrapper.

**Questions:**

- A minor point for clarification on the bibliography: The paper cites both Li et al. (2025a) and Li et al. (2025b), but the corresponding entries in the reference list appear to be identical. Could the authors please clarify the distinction between these two references and their respective contributions to the current work?
- Have the authors considered applying their method to larger-scale Transformer models, such as pretraining a 1B parameter (or larger) generative language model? Such an experiment would allow for a more in-depth exploration of the method's effects on training stability, which is a critical issue in that domain.

---

### Official Review · Reviewer_7KZv · 2025-11-01

**Soundness:** 2
**Presentation:** 2
**Contribution:** 3
**Rating:** 2
**Confidence:** 3

**Summary:**

The paper proposes an initialization scheme that deviates from He-Initialization by scaling layers exponentially with depth (with tunable parameter $\alpha$ applied multiplicatively per-layer). The paper describes a connection to emergence of pattern detection in graphs of compositional functions, following ideas of [Li et al. 2025]. However, the conceptual background remains somewhat vague, arguing with signal propagation vs. noise propagation, to my understanding, assuming different capabilities of handling patterns of varying complexity depending on layer depth. The paper presents experiments with networks with and without shortcuts and with and without batch normalization and observes some practical benefits in terms of accuracy reached, in particular in early training (i.e., the network seems to converge earlier / more quickly).

The paper is a resubmission from an earlier venue (my review is written to be self-contained but I will have to refer to to changes over an earlier version in some tangential aspects below).

**Strengths:**

In principle, the idea of considering the potential patterns emerging in a network of nested functions and trying to adjust the statistics of which/how much learning/processing is performed in different layers sounds highly promising. While I am not aware of concrete results, I would see significant value in trying to study how different levels of compositional depth contribute to building an overall target function in a network.
There are also some interesting measurements, such as Figure 1, which give some insight into the effect of the method. I would be good though to spell out very explicitly which networks / settings were used (Fig. 1b, d), as this has strong impact on signal propagation and training behavior (see below).
The paper also features extensive appendices with additional background information and experiments (I have not been able to go through them all again due to the limited time available and expansive content).

**Weaknesses:**

While I very much like the research question and principle idea of the paper, I see two big issues that still persist in the resubmission:

First, the conceptual background, as far as discussed in the paper, remains very vague. It is not clearly spelled out (and shown by a formal analysis or conclusive experiments) in how far changes in initialization would make "noise" propagation harder than "signal" propagation. A clear definition of noise would be useful, as well as a clear argument how rescaling can distinguish data properties which are one or the other. In general, the whole motivation of the layer rescaling scheme, while sounding very promising in broad strokes, still remains unconcrete. The appendix does give some background, but the main text should make the main ideas clear on a very concrete (preferably even formal) level (as far as I have reread the appendix, the complex concepts from category theory did not help my understanding, but I have only been able to skim over the text in the revision; the original version had not been clear to me).

My second concern is a concrete technical issue. The paper considers, as it should, two major types of architectural variations of a basic DNN: Adding (or not) normalization layers and adding (or not adding) shortcuts. Both of these measures have strong impact on the training behavior of DNNs.

**Batch-normalization:** One first has to note that signal propagation becomes scale-invariant for BN, i.e., what the network computes *does no longer change* under scaling, thus negating any potential effects in terms of signal processing at a fixed point in parameter space. In terms of learning, it actually reverses the workings of a normal network. Because BN does propagate gradients through the normalization step itself, too (i.e., also takes derivatives of gradient contributions), this leads to an inverse gradient scaling: The larger the weights, the smaller the gradients, and scale-invariance implies that actually larger steps are needed for the same effect (one must consider effective rather than absolute learning rates). This means that large weights decrease sensitivity while small weights increase it (an effect not observed in this form in unnormalized networks).

This alone suggest that analysis and experiments of BN and non-BN networks must be separated, as rescaling weights has very different effects. One further issue compounds the problem: BN leads to exploding gradients in lower (near-input) layers; it basically voids the stable gradient propagation of He-Init [Yang et al., ICLR 2019]. Combining this with the reciprocal scaling and scale-invariance leads, non-obviously, to lower layers exploding for one step in weight norm, then freezing and subsequently the whole networks with all layers slowly stabelizing to overall constant effective learning rates. Then, one more issue arises: The convergence time to stability is prohibitively long unless warm-up schemes are used, as the magnitude of the initial step sizes are causal to the "freezing" problem. Thus, BN in truly deep networks requires warm-up (or another fix, see below).

Overall, BN leads to a surprisingly unstable training dynamics, which can be avoided by three (actually almost equivalent, due to scale-invariance) measures: Rescaling layer weights, performing suitable warm-up, or per-layer gradient normalization (such as the LARS-scheduler). The scheme proposed in this paper does exactly the first*), exponential rescaling; so this could be the sole source of the observed positive effect in the BN-regime. The observation that in all BN-cases warm-up comes most closely to the proposed method supports this alternative explanation.

*) see footnote at the end.

However, experiments show that we still show an effect in the case of non-BN networks. Here, the issue must be different in nature. Networks without BN (and no shortcuts, such as VGG used in the paper) are harder to train out of multiple reasons, including potential bias / centering issues (which is not detailed enough to judge; btw: it would be very important to talk about bias intialization and its effect in detail as well in the paper), gradient magnitude excursions at width changes (which He-init cannot handle perfectly), and shrinking dimensionality due to nesting many linear layers that induce a peaked SVD-spectrum. I am not sure where exactly layer rescaling could interfere with the dynamics (I am more familiar with the BN case), but it is conceivable that creating a weighting function as shown in Fig. 1c could restrict training to a subsection of the network, which could be more predictable and thus easier to train (in particular, as training is performed on CIFAR-10, which is not that hard, but VGG is way too big to be trained successfully, which is probably the bigger issue than expressivity). A naive view could be that using small weights makes the network more sensitive to changes, which still have to be propagated through even the down-scaled layers, thereby dominating the effective changes and thus restricting depth implicitly. To not be misunderstood: This is just a guess and probably wrong, but my point is that layer-wise weight rescaling can also in this case have unforeseen numerical effects on which layers are trained that might be the dominant cause (and not differences in emergent expressivity).

**Residual connections:** Residual connections do have a profound impact on trainability as they effectively average paths of different depth, thus reducing all kind of excursions (vanishing / exploding signal, gradients or rank) strongly. As such, rescaling layer by "depth" becomes a very different effective strategy, as one could consider the network an ensemble of subsets of layers of different depth stitched together in an order that is still monotonic in layer index but might skip large parts. The main point is: The motivating thought of controlling how features develop over several layers does not obviously apply here in the same way. As in how far a non-BN version could benefits from layer rescaling is not clear at all to me (so this result in the paper to me personally remains an interesting finding, despite conceptual questions); however, at least for the BN-version, one can again see that warm-up has almost the same effect, which points again to compenstating BN gradient explosions as a natural explanation.

All of the above is obviously debatable and I am not claiming that one could explain all of the observations by already established models; my point of criticism is that one has to look very carefully at the numerical (side-) effects of the network architectures and their interplay with layer (re-)scaling. For the BN case I would think that there are very significant mechanisms at play that are not accounted for in the conceptual considerations in both the introductory as well as experimental part of the paper. For the shortcut aspect, I am at least missing an acknowledgement of the impact, and for the plain, non-BN, non-Res case, it would be at least very useful to track key indicators such as per-layer gradient norms, relative changes to weights, similiarity to initialization etc. to see how training happens over various layers and how this (purely "numerical") behavior is affected by layer-rescaling.

**Summary**

My main point of criticism is (remains) that the paper proposes an interesting new mechanism, but remains vague on the concrete mechanics. At the same time, the effect can be fully explained in at least the BN regime (where I feel knowledgable) by unrelated numerical artifacts in how effective learning rates change over training time due to dynamical artifacts of BN, and I would at least see plausible mechanisms for very different mechanisms also affecting training in the non-BN regime. To exclude that this is the actual issue one should at least monitor the critical effects (response/sensitivity of all of the different layer to training over time). It would be even better to show the hypothesized effect concretely in a controlled experiment.

As I gave a rather low grade, I would also like to emphasize that a rigorous look at numerical side effects is so important because the initial question of trying to capture emergent complexity in compositional networks is so interesting (my rating does not dismiss the idea or direction, rather the opposite); it would be really big if one could find a new effect here; for this reason, it is particularly important to carefully exclude potential alternative explanations issues, as the impact of a positive result could be very significant.


Footnote:
*) An important note concerning the proposed rescaling of layers: On rereading the paper, I noticed that equation 1 and equation 3 actually just impose a monotonic exponential ramp on layer-weights; every next layer is multiplied by a fixed factor. The U-shape only shows up in analyzing (specific, non-BN) concatenations but the scaling is just a linear ramp in the exponent.

**Questions:**

The paper with full appendices is long and complex, and it is quite likely that I have overlooked something important in my arguments above; please correct me accordingly in case of misconceptions. I do not have new/additional questions otherwise.

---

### Official Review · Reviewer_yYBy · 2025-11-03

**Soundness:** 3
**Presentation:** 3
**Contribution:** 3
**Rating:** 6
**Confidence:** 4

**Summary:**

The paper proposes a depth-asymmetric weight initialization scheme that replaces the flat variance assumption of Xavier/Kaiming initialization with a geometric variance ramp across layers. The main idea behind it is to down-scale the first half of the network to suppress input-level noise; up-scale the second half to boost feature-level signals.

**Strengths:**

- The paper provides a novel method for optimization which is well motivated, as the paper has a good motivation section and good related work section.
- The experiments are extensive across different data domains, datasets, and models. This shows clearly the gain of the method.
- The proposes method is also computationally cheap (free as mentioned from the authors) and easy to implement.

**Weaknesses:**

- The paper does not provide any theoretical validation of convergence since this is an optimization method, this should be very important.

**Questions:**

- Are there any theoretical guarantees for why your method works and converges?
- LPVS is not applicable in attention I assume? (line 1025 in the Appendix).

---

### Meta-Review · Area_Chair_Aj1h · 2026-01-06

**Summary:**

This paper proposes LPVS, a depth-asymmetric weight initialization scheme that rescales standard initializations (e.g., He/Xavier) exponentially with layer depth, with the goal of suppressing input-level noise in shallow layers while amplifying feature-level signals in deeper layers. Reviewers generally found the idea interesting, easy to implement, and empirically effective, with experiments across multiple architectures, datasets, and training regimes showing faster convergence and moderate accuracy improvements. However, despite these positive signals, the reviews converge on the assessment that the mechanistic understanding and conceptual grounding of the method remain insufficiently developed, particularly in light of strong confounding effects introduced by batch normalization and residual connections.

**Reviewer Concerns:**

The reviewers acknowledge several strengths: the research question is well motivated, the method is simple and computationally free, and the experimental evaluation is broad. Reviewer yYBy found the motivation and empirical results convincing overall, and Reviewer AJWU noted that the method integrates seamlessly into existing pipelines and can alleviate issues such as dead ReLUs. However, substantial concerns remain unresolved.

The most significant and detailed critique, articulated by Reviewer 7KZv, is that the paper’s conceptual explanation is vague and not supported by sufficiently controlled analysis or experiments. In particular, the claimed distinction between “signal” and “noise” propagation is not clearly defined, nor is it formally or empirically demonstrated that the proposed rescaling selectively affects one more than the other. For networks with batch normalization, the reviewer argues convincingly that the observed benefits can plausibly be explained by well-known BN-related dynamics—such as scale invariance, inverse gradient scaling, exploding gradients in early layers, and the need for warm-up—rather than by a new representational or compositional effect. Indeed, the close alignment between LPVS and warm-up schemes in BN settings strongly supports this alternative explanation.

**Reviewer Scores:**

Reviewer 7KZv (score: 2): This reviewer provided an extensive and technically detailed critique, arguing that the main effects can be explained by known BN-related numerical artifacts and that the paper does not adequately rule out these alternatives. The rebuttal did not resolve these concerns, so the score would remain at 2.

Reviewer AJWU (score: 4): While recognizing the novelty and empirical benefits of LPVS, this reviewer raised concerns about unclear presentation, insufficient discussion of stability issues (e.g., warm-up, normalization), and missing experimental details. These are partially addressable but central enough that the score would remain at 4.

Reviewer yYBy (score: 6): This reviewer viewed the method as well motivated, broadly evaluated, and easy to apply, but explicitly noted the lack of theoretical guarantees and stated they “would not mind if the paper is rejected.” Given the unresolved conceptual issues raised by others, the score would likely remain at 6 rather than increase.

---

### Decision · Program_Chairs · 2026-01-26

Reject